# 3D-GP-LMVIC: Learning-based Multi-View Image Compression with 3D Gaussian Geometric Priors

## Abstract

Multi-view image compression is vital for 3D-related applications. Existing methods often rely on 2D projection similarities between views to estimate disparity, performing well with small disparities, such as in stereo images, but struggling with more complex disparities from wide-baseline setups, common in virtual reality and autonomous driving systems. To overcome this limitation, we propose a novel approach: learning-based multi-view image compression with 3D Gaussian geometric priors (3D-GP-LMVIC). Our method leverages 3D Gaussian Splatting to derive geometric priors of the 3D scene, enabling more accurate disparity estimation between views within the compression model. Additionally, we introduce a depth map compression model to reduce redundancy in geometric information across views. A multi-view sequence ordering method is also proposed to enhance correlations between adjacent views. Experimental results demonstrate that 3D-GP-LMVIC surpasses both traditional and learning-based methods in performance, while maintaining fast encoding and decoding speed. The code is available at https://anonymous.4open.science/r/3D-GP-LMVIC-8FFA.

## 1 Introductioin

The rapid advancement of 3D applications has led to an explosion of multi-view image data across various fields, including virtual reality (VR) (Anthes et al., 2016), augmented reality (AR) (Schmalstieg & Hollerer, 2016), visual simultaneous localization and mapping (vSLAM) (Mokssit et al., 2023), 3D scene understanding (Dai et al., 2017), autonomous driving (Chen et al., 2017), and medical imaging (Hosseinian & Arefi, 2015). In particular, applications like VR and AR, which rely on high-quality multi-view visual content to create immersive experiences, generate a massive volume of data that poses significant challenges for storage and transmission. This makes the development of efficient compression techniques crucial for managing the increasing data demands in these fields.

Current multi-view coding standards, such as H.264-based MVC (Vetro et al., 2011) and H.265-based MV-HEVC (Hannuksela et al., 2015), have been developed to compress multi-view media by extending their respective base standards and exploiting redundancies across multiple views. These standards incorporate inter-view prediction, where blocks in one view are predicted based on corresponding blocks in neighboring view. Disparity estimation is employed to calculate positional differences of objects between views, aiding in the prediction of pixel values. However, these methods rely on manually designed modules, limiting the system's ability to fully leverage end-to-end optimization. Consequently, the spatial correlation among views may not be fully exploited, leading to suboptimal compression performance.

Learning-based single image compression has seen remarkable advancements (Ballé et al., 2017; 2018; Minnen et al., 2018), inspiring extensions of these methods to multi-view image coding (Deng et al., 2021; Lei et al., 2022; Zhang et al., 2023; Liu et al., 2024). A central challenge in these extensions lies in the accurate estimation of disparities across different views. For example, Deng et al. (2021; 2023) employ a simple 3x3 homography matrix for disparity estimation, which, while efficient, struggles with complex scene disparities. Alternatively, Ayzik & Avidan (2020); Huang et al. (2023) utilize patch matching method to align the reference view with the target view. This approach is effective for horizontal or vertical view shifts but falls short when addressing non-rigid

deformations caused by view rotations. Similarly, Zhai et al. (2022) assume that disparity occurs only along the horizontal axis in their stereo matching method, which suffices for stereo images but is inadequate for more complex view transformations where disparity is not limited to the horizontal axis. Some methods leverage cross-attention mechanisms for implicit alignment (Wödlinger et al., 2022; Zhang et al., 2023; Liu et al., 2024). For instance, Zhang et al. (2023) enhance the target view's representation by multiplying its query with the reference view's key and value, effectively incorporating reference view features into the target view. However, these methods primarily establish correlations between two views by 2D projection similarities, without considering the 3D spatial relationships between the views and the captured objects.

In addition, we are impressed by 3D Gaussian Splatting (3D-GS) (Kerbl et al., 2023), a novel technique for generating 3D representations from multi-view images and synthesizing novel views. This method achieves remarkable results by representing scenes as mixtures of small, colored Gaussians, capturing intricate geometric details and continuous depth and texture variations. The probabilistic nature of Gaussians allows for smooth, accurate modeling of complex surfaces, enabling the creation of realistic and detailed 3D environments. Additionally, the fast differentiable renderer inherent in 3D Gaussian Splatting ensures efficient real-time inference.

Building on prior investigation, we propose a novel learning-based multi-view image compression framework with 3D Gaussian geometric priors (3D-GP-LMVIC), which employs 3D-GS as a geometric prior to guide disparity estimation between views. Specifically, 3D-GS generates a depth map for each view, providing precise spatial information at the pixel level. This enables accurate correspondence between views, allowing the compression model to effectively fuse features from reference views. Due to positional and angular disparities between views, images generally do not fully overlap, and merging non-overlapping regions may introduce noise. To address this, we design a mask based on the 3D Gaussian geometric prior to identify overlapping regions, ensuring more accurate feature fusion. Additionally, since depth maps are required during decoding, we propose a depth map compression model to efficiently reduce geometric redundancy across views, incorporating a cross-view depth prediction module to capture inter-view geometric correlations. Finally, recognizing the importance of field of view (FoV) overlap in redundancy reduction, we introduce a multi-view sequence ordering method to address the issue of low overlap between adjacent views in unordered sequences. This method defines a distance measure between view pairs to guide the ordering of view sequences.

- We propose a learning-based multi-view image compression framework with 3D Gaussian geometric priors (3D-GP-LMVIC), which utilizes 3D Gaussian geometric priors for precise disparity estimation between views, thereby enhancing multi-view image compression efficiency. Additionally, we design a mask based on these priors to identify overlapping regions between views, effectively guiding the model to retain useful cross-view information.

- We also present a depth map compression model aimed at reducing geometric redundancy across views. Furthermore, a multi-view sequence ordering method is proposed to improve the correlation between adjacent views.

- Experimental results show that our framework surpasses both traditional and learning-based multi-view image coding methods in compression efficiency, while also providing fast encoding and decoding speed. Moreover, our disparity estimation method demonstrates greater visual accuracy compared to existing two-view disparity estimation methods.

## 2 RELATED WORKS

**Single Image Coding.** Traditional image codecs, such as JPEG (Wallace, 1992), BPG (Bellard, 2014), and VVC (Bross et al., 2021), employ manually designed modules like DCT, block-based coding, and quadtree plus binary tree partitioning to balance compression and visual quality. These methods, however, do not achieve end-to-end joint optimization, limiting their performance.

In recent years, learning-based image compression methods have integrated autoencoders with differentiable entropy models to enable end-to-end optimization of rate-distortion loss. Early works, such as Ballé et al. (2017; 2018), introduced generalized divisive normalization (GDN) (Ballé et al.,

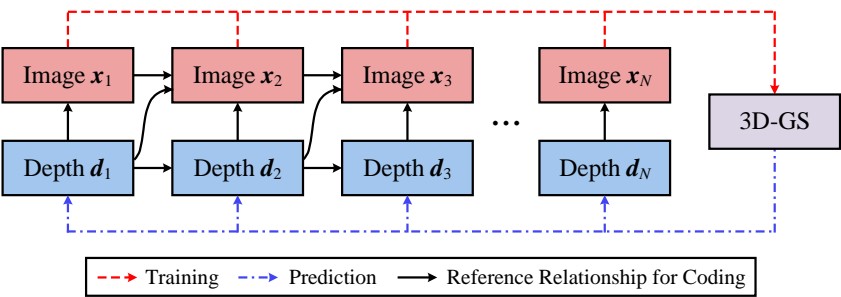

Figure 1: The overall pipeline of 3D-GP-LMVIC

2016) and proposed factorized and hyperprior entropy models. Subsequent research (Minnen et al., 2018; He et al., 2021; Jiang et al., 2023) incorporated autoregressive structures into entropy models, resulting in more accurate probability predictions. These advancements have laid the foundation for learning-based multi-view image coding.

**Multi-view Image Coding.** Traditional multi-view image codecs, such as MVC (Vetro et al., 2011) and MV-HEVC (Hannuksela et al., 2015), extend H.264 and H.265, respectively, by incorporating inter-view correlation modeling to eliminate redundant information between different views. However, these modules are manually designed, potentially limiting their ability to fully exploit cross-view information.

Learning-based multi-view image coding primarily focuses on stereo image coding (Deng et al., 2021; Lei et al., 2022; Wödlinger et al., 2022; Zhai et al., 2022; Deng et al., 2023; Liu et al., 2024) and distributed image coding (Ayzik & Avidan, 2020; Huang et al., 2023; Zhang et al., 2023). These methods either rely on finding explicit pixel coordinate correspondences between views or use attention-based implicit correspondence modeling to capture inter-view correlations. However, they model inter-view correlations based solely on two-dimensional view images, which may not fully reflect the correspondences in the original three-dimensional space.

**3D Gaussian Splatting.** 3D Gaussian Splatting (Kerbl et al., 2023; Hamdi et al., 2024) introduces a differentiable point-based rendering technique that represents 3D points as Gaussian functions (mean, variance, opacity, color) and projects these 3D Gaussians onto a view to form an image. This differentiable point-based rendering function allows for the backward update of the attributes of the 3D Gaussians, ensuring that their geometrical and textural properties match the original 3D scene. This approach inspired us to utilize 3D Gaussian Splatting to obtain geometric priors of the original 3D scene, aiding in the task of multi-view image compression.

## 3 PROPOSED METHOD

Figure 1 shows the overall pipeline of 3D-GP-LMVIC. Given a set of multi-view image sequences $\mathcal{X} = \{x_1, x_2, x_3, \cdots, x_N\}$, a 3D-GS is trained to estimate depth map $d_n$ for each image $x_n$. Both $x_n$ and $d_n$ are compressed, with the coding reference relationships indicated by black solid arrows in Figure 1. Prior to compressing the image $x_n$, it is necessary to compress $x_{n-1}$, $d_{n-1}$, and $d_n$. The disparity relationship between the $(n-1)$-th and $n$-th views is inferred from the reconstructed depth maps $\hat{d}_{n-1}$ and $\hat{d}_n$. Subsequently, based on the estimated disparity relationship, as well as the intermediate features and the reconstructed image $\hat{x}_{n-1}$ obtained during the image decoding process of the $(n-1)$-th view, $x_n$ is compressed. When compressing the depth map $d_n$, the model employs the predicted depth map derived from $\hat{d}_{n-1}$ as a reference. The same neural network architecture and model parameters are used consistently across all views for both image compression and depth map compression. For the initial view lacking reference information, full-zero tensors are provided as the input reference.

The remainder of this section is structured as follows: Section 3.1 elaborates on the method for depth map estimation for a given view using the 3D-GS and the estimation of inter-view disparities. Section 3.2 discusses the compression model for images and depth maps. Section 3.3 introduces

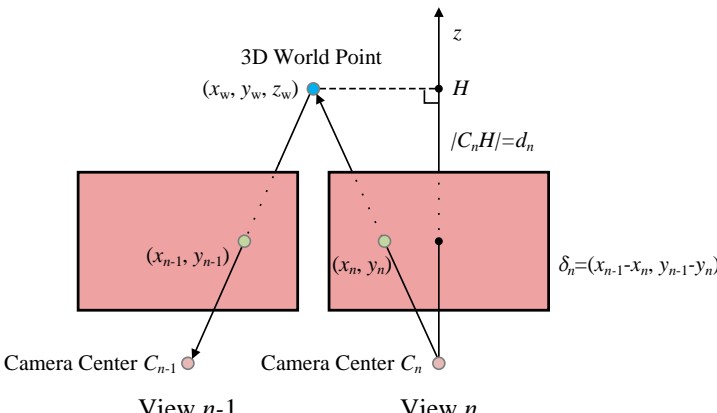

Figure 2: Illustration of depth-based disparity estimation

a multi-view sequence ordering method designed to ensure sufficient overlap of captured objects between adjacent views.

### 3.1 3D-GS BASED DEPTH AND DISPARITY ESTIMATION

For an image $x_n \in \mathbb{R}^{W \times H \times 3}$ with spatial dimensions $W$ and $H$, we aim to derive a depth map $d_n \in \mathbb{R}^{W \times H}$, representing the $z$-axis coordinates of each pixel's corresponding scene points in the camera coordinate system. This depth map facilitates the estimation of disparities between different views.

In the context of the 3D-GS framework, consider a set of $M$ ordered 3D points projected along a ray from the camera through a pixel. The rendered pixel color $c$ can be expressed as:

$$c = \sum_{i=1}^{M} T_i \alpha_i c_i, \quad \text{with } T_i = \prod_{j=1}^{i-1} (1 - \alpha_j). \tag{1}$$

Here, $c_i$ and $\alpha_i$ represent the color and opacity (density) of the point, respectively, derived from the point's 3D Gaussian properties. The factor $T_i$ denotes the transmittance along the ray, indicating the fraction of light reaching the camera without being occluded.

In Eq. 1, $T_i$ serves as a weight for the contribution of each point's color to the pixel's final color, diminishing from 1 to 0 as $i$ increases due to cumulative absorption. To estimate the depth of the pixel $d$, we use the depth $z_i$ of the first point where $T_i$ drops below 0.5, following the median depth estimation approach outlined in Luiten et al. (2024):

$$d = z_{i^*}, \quad \text{where } i^* = \min\{i \mid T_i < 0.5\}. \tag{2}$$

It is worth noting that the original 3D-GS (Kerbl et al., 2023) employs a weighted averaging approach, using $T_i\alpha_i$ as the weight for each 3D Gaussian along the ray to compute depth. In contrast, alignment experiments in Appendix E demonstrate that the median depth estimation approach achieves better alignment performance.

Next, we aim to estimate the disparity $\mathbf{\Delta}_n \in \mathbb{R}^{W \times H \times 2}$ between views based on the estimated depth map. This disparity represents the pixel-wise shift of each scene point's projection across different views. Disparity estimation captures the geometric relationships between views, facilitating the modeling of inter-view correlations.

Figure 2 illustrates the depth-based disparity estimation. To estimate the disparity, a pixel $(x_n, y_n)$ in the $n$-th view is back-projected into 3D space using the depth $d_n$ to obtain the world coordinates $(x_w, y_w, z_w)$. This 3D world point is then projected into the $(n-1)$-th view to obtain the corresponding pixel coordinates $(x_{n-1}, y_{n-1})$. The transformations involved are as follows:

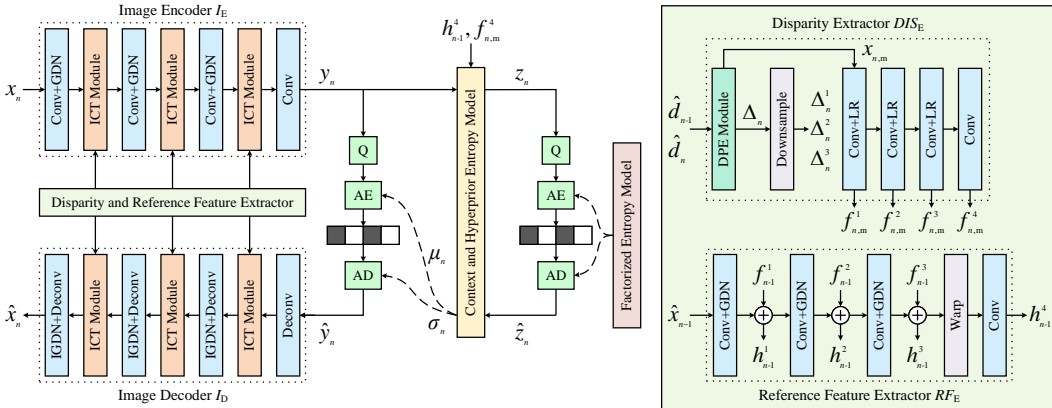

Figure 3: The architecture of the proposed image compression model. 'LR' represents the Leaky ReLU activation function, 'Q' denotes the quantization operation, and 'AE'/'AD' refer to the arithmetic encoder/decoder, respectively.

$$\begin{bmatrix} x_{\mathrm{w}} \\ y_{\mathrm{w}} \\ z_{\mathrm{w}} \\ 1 \end{bmatrix} = V_n^{-1} \cdot \mathrm{aug}\left( K^{-1} d_n \begin{bmatrix} x_n \\ y_n \\ 1 \end{bmatrix} \right), \quad \text{with } \mathrm{aug}\left( \begin{bmatrix} x \\ y \\ z \end{bmatrix} \right) = \begin{bmatrix} x \\ y \\ z \\ 1 \end{bmatrix},$$

$$d'_{n-1} \begin{bmatrix} x_{n-1} \\ y_{n-1} \\ 1 \end{bmatrix} = K \cdot \mathrm{deaug}\left( V_{n-1} \begin{bmatrix} x_{\mathrm{w}} \\ y_{\mathrm{w}} \\ z_{\mathrm{w}} \\ 1 \end{bmatrix} \right), \quad \text{with } \mathrm{deaug}\left( \begin{bmatrix} x \\ y \\ z \\ 1 \end{bmatrix} \right) = \begin{bmatrix} x \\ y \\ z \end{bmatrix},$$

(3)

where $K \in \mathbb{R}^{3 \times 3}$ denotes the camera intrinsic matrix, and $V_n, V_{n-1} \in \mathbb{R}^{4 \times 4}$ are the extrinsic matrices corresponding to the $n$-th and $(n-1)$-th views, respectively. The camera parameters are calibrated using SfM (Schonberger & Frahm, 2016). $d'_{n-1}$ represents the depth of the 3D world point in the camera coordinate system of the $(n-1)$-th view. The resulting disparity $\delta_n = (x_{n-1} - x_n, y_{n-1} - y_n)$ for each pixel is then compiled into the disparity map $\mathbf{\Delta}_n$.

Finally, we define a mask $\boldsymbol{x}_{n,\mathrm{m}} \in \mathbb{R}^{W \times H}$ that identifies whether each pixel in the $n$-th view maps to the same 3D world point as the corresponding pixel in the $(n-1)$-th view through disparity estimation. The mask's criteria are: (1) the projected pixel must reside within the valid image region in the $(n-1)$-th view, (2) the corresponding 3D world point must be in the positive $z$-half-space of the $(n-1)$-th view's coordinate system, and (3) no occlusion must exist along the line of sight, i.e., $d'_{n-1}$ from Eq. 3 must be less than the estimated depth along the ray in the $(n-1)$-th view. This can be formulated as:

$$\boldsymbol{x}_{n,\mathrm{m}}[i,j] = \begin{cases} 1 & \text{if } 0 < \mathbf{\Delta}_n[i,j,0] + i + 0.5 < W \\ & \text{and } 0 < \mathbf{\Delta}_n[i,j,1] + j + 0.5 < H \\ & \text{and } 0 < \boldsymbol{d}'_{n-1}[i,j] < \mathrm{Warp}(\boldsymbol{d}_{n-1}, \mathbf{\Delta}_n)[i,j], \\ 0 & \text{otherwise}, \end{cases}$$

(4)

where $\boldsymbol{d}'_{n-1} \in \mathbb{R}^{W \times H}$ represents the tensor containing the depth values $d'_{n-1}$ for each pixel, and $\mathrm{Warp}(\cdot, \cdot)$ denotes the warping operation based on the given disparity. Appendix A outlines the algorithmic process for disparity and mask estimation.

## 3.2 COMPRESSION FRAMEWORK FOR IMAGES AND DEPTH MAPS

### 3.2.1 IMAGE COMPRESSION MODEL

As shown in Figure 3, the disparity extractor $DIS_{\mathrm{E}}$ utilizes reconstructed depth maps $\hat{\boldsymbol{d}}_{n-1}$ and $\hat{\boldsymbol{d}}_n$ to extract multi-scale disparities and feature masks. The reference feature extractor $RF_{\mathrm{E}}$ gen-

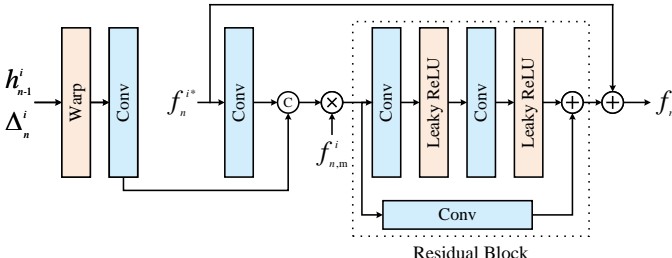

Figure 4: Illustration of the proposed image context transfer module at the $i$-th scale.

erates multi-scale reference features from the reconstructed image $\hat{x}_{n-1}$ and intermediate features $\{f_{n-1}^i \mid i = 1, 2, 3\}$ derived during $\hat{x}_{n-1}$'s reconstruction. Subsequently, the image encoder $I_{\mathrm{E}}$ and decoder $I_{\mathrm{D}}$ incorporate the reference features, aligned using the extracted disparities, into the backbone network. This process is formalized as:

$$y_n = I_{\mathrm{E}}(x_n, DIS_{\mathrm{E}}(\hat{d}_{n-1}, \hat{d}_n), RF_{\mathrm{E}}(\hat{x}_{n-1}, \{f_{n-1}^i \mid i = 1, 2, 3\})),$$
$$\hat{y}_n = Q(y_n), \tag{5}$$
$$\hat{x}_n = I_{\mathrm{D}}(\hat{y}_n, DIS_{\mathrm{E}}(\hat{d}_{n-1}, \hat{d}_n), RF_{\mathrm{E}}(\hat{x}_{n-1}, \{f_{n-1}^i \mid i = 1, 2, 3\})).$$

For entropy coding, we utilize the hyperprior entropy model (Ballé et al., 2018) and the quadtree partition-based entropy model (QPEM) (Li et al., 2023). The hyperprior entropy model transforms $y_n$ into a hyperprior representation $z_n$. The quantized hyperprior representation $\hat{z}_n$ is then used to accurately model the probability distribution of $\hat{y}_n$. The conditional probability distribution $p_{\hat{y}_n \mid \hat{z}_n}$ is defined as:

$$p_{\hat{y}_n \mid \hat{z}_n}(\hat{y}_n \mid \hat{z}_n) \sim \mathcal{N}(\mu_n, \sigma_n^2). \tag{6}$$

Additionally, QPEM enhances entropy coding efficiency by introducing diverse spatial contexts while maintaining high coding speed.

**Disparity extractor.** As illustrated in Figure 3, we firstly employ the disparity estimation (DPE) module to derive the disparity map $\Delta_n$ and the corresponding mask $x_{n,\mathrm{m}}$, following the method outlined in Section 3.1, using $\hat{d}_{n-1}$ and $\hat{d}_n$. Subsequently, $\Delta_n$ undergoes a series of downsampling operations to produce multi-scale disparity maps $\{\Delta_n^i \mid i = 1, 2, 3\}$, which will facilitate multi-scale feature alignment. The mask $x_{n,\mathrm{m}}$ is further processed by the disparity mask extractor to extract feature masks $\{f_{n,\mathrm{m}}^i \mid i = 1, 2, 3, 4\}$.

**Reference feature extractor.** The referenc, aiding in the extraction of relevant features.e feature extractor takes as inputs $\hat{x}_{n-1}$, $\{f_{n-1}^i \mid i = 1, 2, 3\}$, and $\Delta_n^3$ to extract multi-scale reference features $\{h_{n-1}^i \mid i = 1, 2, 3, 4\}$, as shown in Figure 3. Specifically, $h_{n-1}^4$ is derived by aligning $h_{n-1}^3$ with $\Delta_n^3$ and subsequently applying a convolutional layer.

**Image context transfer module.** To incorporate the reference feature $\{h_{n-1}^i \mid i = 1, 2, 3\}$ obtained from the $(n-1)$-th view into the image backbone encoder and decoder, enhancing feature representation, we introduce the image context transfer (ICT) module. As depicted in Figure 4, the input feature $f_n^{i^*}$ from the backbone network and the aligned feature of $h_{n-1}^i$ via $\Delta_n^i$ are concatenated after a convolutional layer. The concatenated feature is then element-wise multiplied with $f_{n,\mathrm{m}}^i$, followed by a residual block for feature refinement. Multiplication with the feature masks filters relevant information from the reference features. The final step involves element-wise addition of this refined feature to $f_n^{i^*}$, yielding the output feature $f_n^i$.

### 3.2.2 DEPTH MAP COMPRESSION MODEL

The compression and decompression of the depth map $d_n$ leverage $\hat{d}_{n-1}$ as a reference. Initially, $\hat{d}_{n-1}$ is processed by the depth prediction extractor $DEP_{\mathrm{E}}$, which generates multi-scale depth prediction features and corresponding feature masks. Subsequently, the depth encoder $D_{\mathrm{E}}$ and decoder $D_{\mathrm{D}}$ integrate these extracted features and masks into the backbone network. This process is formal-

ized as:

$$
\begin{aligned}
\boldsymbol{y}_{d_n} &= D_{\mathrm{E}}(\boldsymbol{d}_n, DEP_{\mathrm{E}}(\hat{\boldsymbol{d}}_{n-1})), \\
\hat{\boldsymbol{y}}_{d_n} &= Q(\boldsymbol{y}_{d_n}), \\
\hat{\boldsymbol{d}}_n &= D_{\mathrm{D}}(\hat{\boldsymbol{y}}_{d_n}, DEP_{\mathrm{E}}(\hat{\boldsymbol{d}}_{n-1})).
\end{aligned}
\tag{7}
$$

The entropy coding scheme incorporates both the hyperprior entropy model and the QPEM. The latent representation $\boldsymbol{y}_{d_n}$ is transformed into a hyperprior representation $\boldsymbol{z}_{d_n}$ using the hyperprior entropy model. Similar to the image compression model, the quantized hyperprior representation $\hat{\boldsymbol{z}}_{d_n}$ is used to model the probability distribution of $\hat{\boldsymbol{y}}_{d_n}$. Additional details about the depth map compression model are provided in Appendix B.

### 3.2.3 Training Loss

For each training step, a randomly selected subsequence of length 4 from a multi-view sequence serves as the training sample. The training loss comprises the distortion losses for both the reconstructed image and depth map, as well as the estimated compression rates for the encoded image and depth map, for each view in the training sample:

$$
L = \sum_{n=s}^{s+3} w_{n-s+1} \left[ \lambda_{\mathrm{img}} D(\boldsymbol{x}_n, \hat{\boldsymbol{x}}_n) + \lambda_{\mathrm{dep}} \mathrm{MSE}(\boldsymbol{d}_n, \hat{\boldsymbol{d}}_n) + R(\hat{\boldsymbol{y}}_n) + R(\hat{\boldsymbol{z}}_n) + R(\hat{\boldsymbol{y}}_{d_n}) + R(\hat{\boldsymbol{z}}_{d_n}) \right],
\tag{8}
$$

where $D(\cdot, \cdot)$ denotes the distortion, $\mathrm{MSE}(\cdot, \cdot)$ represents the mean squared error (MSE), and $R(\cdot)$ indicates the estimated compression rates. The hyperparameters $\lambda_{\mathrm{img}}$ and $\lambda_{\mathrm{dep}}$ control the contributions of the image and depth map distortion losses, respectively. The weights $\{w_i \mid i = 1, 2, 3, 4\}$, as referenced from Li et al. (2023), adjust the influence of each view on the overall training loss.

### 3.3 Multi-View Sequence Ordering

Given the significant impact of FoV overlap between adjacent views on inter-view correlations, we propose a multi-view sequence ordering method to alleviate the issue of insufficient overlap in unordered sequences. We define a distance metric to evaluate inter-view overlap and employ a greedy algorithm to find an improved sequence.

In Eq. 3, if $V_{n-1}V_n^{-1} = I$, then $(x_n, y_n) = (x_{n-1}, y_{n-1})$. This indicates that each pixel in the $n$-th view lies within the valid image area of the $(n-1)$-th view, indicating high overlap. Thus, for any two views $i$ and $j$, we measure overlap by the proximity of $V_i V_j^{-1}$ to the identity matrix:

$$
\mathcal{D}_{\mathcal{V}}(i, j) = \|V_i V_j^{-1} - I\|.
\tag{9}
$$

Appendix C demonstrates that $\mathcal{D}_{\mathcal{V}}(i, j)$ is a distance metric for both the 2-norm and Frobenius norm. The Frobenius norm is utilized in our experiments. After determining pairwise distances, a greedy algorithm is employed, starting from an initial sequence with only one view, iteratively selecting the view closest to the last view in the sequence.

## 4 Experiments

### 4.1 Experimental Setup

**Datasets.** We evaluate our model on three multi-view image datasets: Tanks&Temples (Knapitsch et al., 2017), Mip-NeRF 360 (Barron et al., 2022), and Deep Blending (Hedman et al., 2018). These datasets feature a wide variety of indoor and outdoor scenes, each containing dozens to over a thousand images captured from different views. Further details on the datasets are provided in Appendix D.

**Benchmarks.** We compare our approach against several baselines, including traditional multi-view codec: MV-HEVC (Hannuksela et al., 2015); learning-based multi-view image codecs: two variants

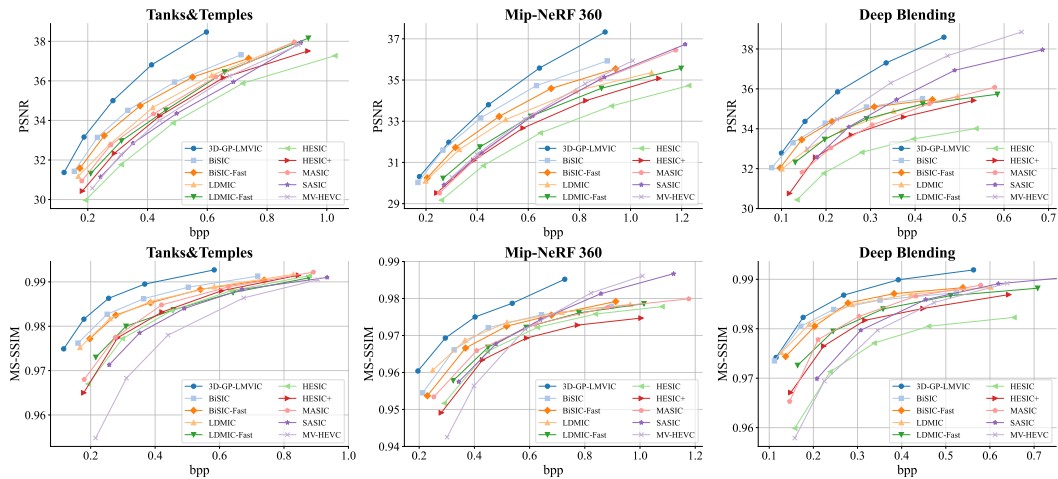

Figure 5: Rate-distortion curves of the proposed method compared with baselines.

Table 1: BDBR comparison of different methods relative to MV-HEVC.

| Methods | Tanks&Temples | | Mip-NeRF 360 | | Deep Blending | |
|---------|------|---------|------|---------|------|---------|
| | PSNR | MS-SSIM | PSNR | MS-SSIM | PSNR | MS-SSIM |
| HAC | 636.81% | 350.72% | 374.20% | 294.42% | 673.57% | 418.85% |
| HESIC | 12.66% | -26.29% | 28.41% | -6.18% | 85.38% | 3.91% |
| HESIC+ | -4.85% | -30.42% | 9.48% | -5.11% | 32.5% | -19.14% |
| MASIC | -12.57% | -34.19% | 3.26% | -9.11% | 43.6% | -9.33% |
| SASIC | 3.39% | -18.59% | 2.40% | -3.70% | 24.64% | -9.48% |
| LDMIC-Fast | -8.56% | -27.76% | 1.72% | -6.21% | 24.25% | -23.31% |
| LDMIC | -16.27% | -44.33% | -13.12% | -25.39% | 16.88% | -41.94% |
| BiSIC-Fast | -26.59% | -42.93% | -20.61% | -23.23% | -8.24% | -41.80% |
| BiSIC | -30.89% | -49.96% | -29.87% | -30.75% | -15.46% | -48.47% |
| 3D-GP-LMVIC | **-47.48%** | **-63.69%** | **-34.69%** | **-40.25%** | **-27.31%** | **-54.15%** |

of HESIC (Deng et al., 2021), MASIC (Deng et al., 2023), SASIC (Wödlinger et al., 2022), two variants of LDMIC (Zhang et al., 2023), and two variants of BiSIC (Liu et al., 2024); as well as the 3D-GS compression method: HAC (Chen et al., 2024). Further details on the baseline configurations are provided in Appendix D.

**Metrics.** Image reconstruction quality is measured using peak signal-to-noise ratio (PSNR) and multi-scale structural similarity index (MS-SSIM) (Wang et al., 2003). Bitrate is expressed in bits per pixel (bpp). In addition to plotting RD curves, the Bjøntegaard Delta bitrate (BDBR) is calculated to quantify the average bitrate savings across varying reconstruction qualities. Lower BDBR values indicate better performance.

**Implementation Details.** The model was trained using five different configurations of $(\lambda_{\text{img}}, \lambda_{\text{dep}})$: $((256, 64), (512, 128), (1024, 128), (2048, 128), (4096, 128))$ when the image distortion loss is MSE, and $((8, 64), (16, 128), (32, 128), (64, 128), (128, 128))$ when using MS-SSIM. The weights $w_i$ for four consecutive views were set to $(0.5, 1.2, 0.5, 0.9)$. The model was trained for 300 epochs with an initial learning rate of $10^{-4}$, which was progressively decayed by a factor of 0.5 every 60 epochs.

## 4.2 EXPERIMENTAL RESULTS

**Coding performance.** Figure 5 presents the rate-distortion curves of the compared methods, while Table 1 summarizes the BDBR of each codec relative to MV-HEVC. Across the three datasets, the proposed 3D-GP-LMVIC consistently outperforms the baselines in both PSNR and MS-SSIM, demonstrating its effectiveness in reducing inter-view redundancy. For instance, on the Tanks&Temples dataset, 3D-GP-LMVIC achieves a BDBR reduction of 16.59% for PSNR and

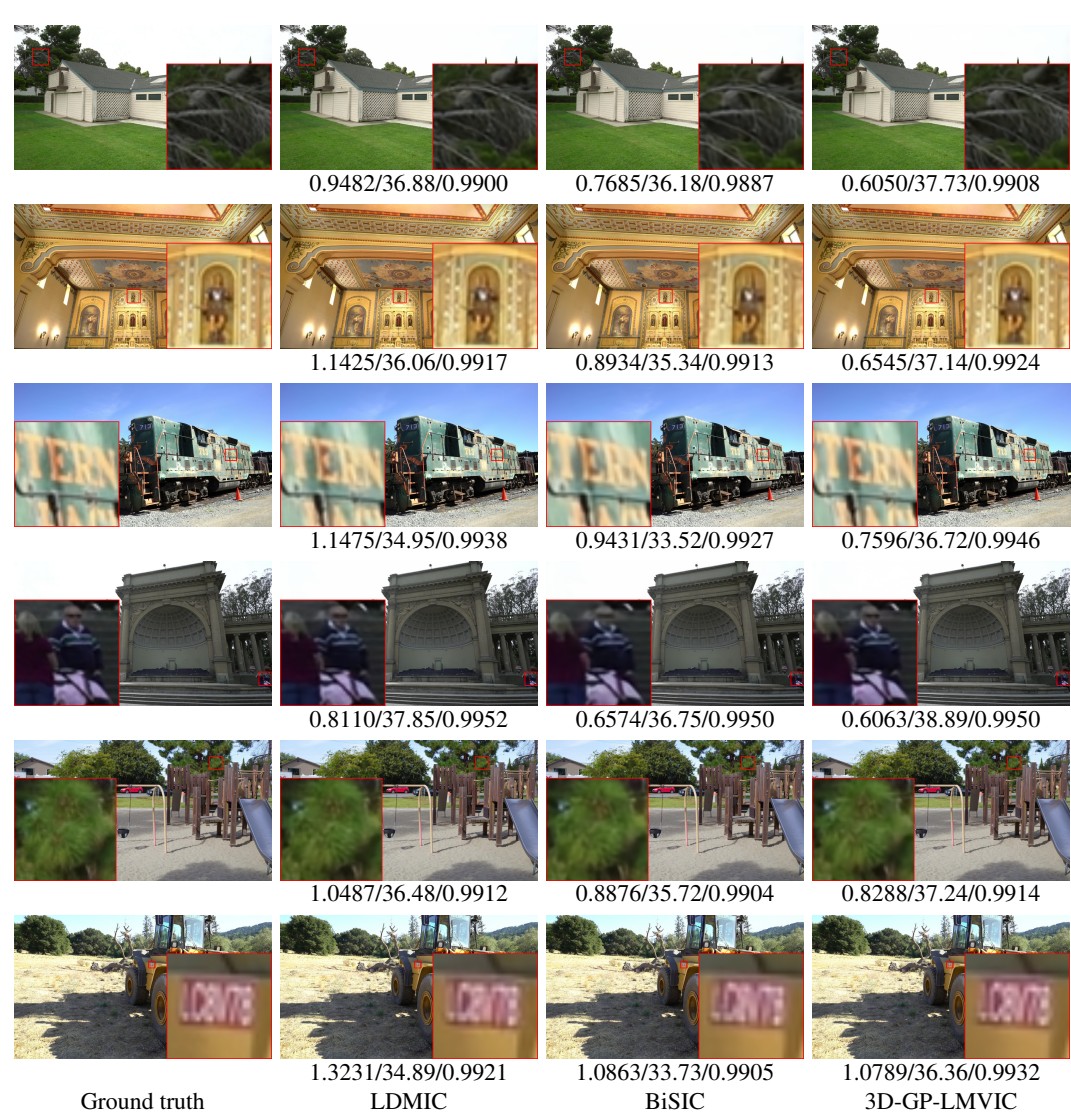

Figure 6: Visual Comparison of LDMIC, BiSIC, and 3D-GP-LMVIC on the Tanks&Temples Dataset. Compression performance is reported as bpp/PSNR/MS-SSIM.

13.73% for MS-SSIM compared to BiSIC. The BDBR of HAC is relatively higher, likely due to the inclusion of 3D scene information in addition to 2D image representations. Appendix F provides supplementary experiments on coding performance.

**Visualization.** In Figure 6, we present examples from the Tanks&Temples dataset to visually compare the performance of LDMIC, BiSIC, and 3D-GP-LMVIC. The results demonstrate that 3D-GP-LMVIC preserves more texture details and achieves higher reconstruction quality for elements like branches, humans, and text, while consuming fewer bits.

**Encoding and decoding time comparison.** Table 2 presents the encoding and decoding runtimes of six learning-based image codecs, evaluated on a platform equipped with an Intel(R) Xeon(R) Gold 6330 CPU @ 2.00GHz and an NVIDIA RTX A6000 GPU. The neural network components of the codecs were executed on the GPU, while the entropy coding was handled by the CPU. 3D-GP-LMVIC achieved encoding and decoding times of 0.19s and 0.18s, respectively, making it one of the faster methods. This speed can be attributed to its relatively simple network structure and the QPEM, which balances coding speed and compression efficiency. Both SASIC and LDMIC-Fast showed fast runtimes due to their use of highly parallelizable hyperprior and checkerboard entropy models, respectively. Besides, HESIC+, MASIC, and LDMIC utilized autoregressive entropy models, resulting in comparatively slower runtimes.

Table 2: Average encoding and decoding runtime of learning-based codecs on Tanks&Temples (image resolution: $978 \times 546$).

| Operation | Codecs | | | | | |
|---|---|---|---|---|---|---|
| | HESIC+ | MASIC | SASIC | LDMIC-Fast | LDMIC | 3D-GP-LMVIC |
| Encoding | 4.35s | 4.38s | 0.06s | 0.11s | 4.24s | 0.19s |
| Decoding | 10.73s | 10.78s | 0.09s | 0.09s | 10.63s | 0.18s |

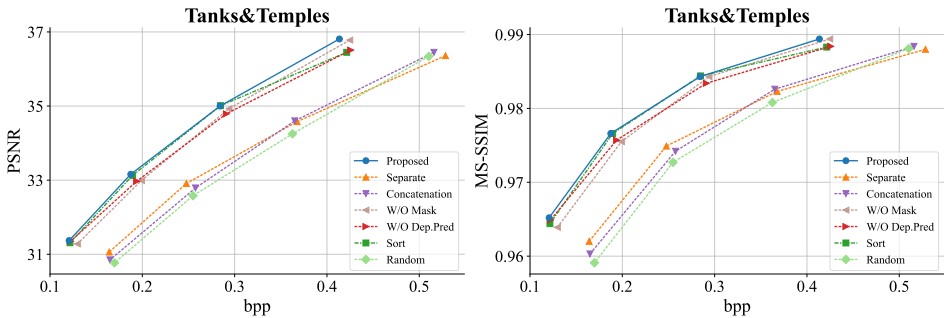

Figure 7: Rate-distortion curves of different ablation baselines on the Tanks&Temples dataset.

## 4.3 ABLATION STUDY

**Codec components.** To assess the contribution of codec components, we performed ablation experiments on the Tanks&Temples dataset. The rate-distortion curves are shown in Figure 7. Specifically, we evaluated the following baselines: (1) *Separate*: encoding and decoding without cross-view information; (2) *Concatenation*: direct feature concatenation from reference view without alignment; (3) *W/O Mask*: removal of both image and depth mask; (4) *W/O Dep.Pred*: excluding depth prediction in the depth map compression model. These baselines resulted in bitrate increases of 41.07% (44.24%), 42.75% (47.52%), 7.19% (8.47%), and 8.03% (8.02%) for PSNR (MS-SSIM), respectively, compared to the proposed method. The experimental results validate the effectiveness of the proposed components. Appendix E provides further analysis on alignment.

**Multi-view sequence ordering.** As illustrated in Figure 7, we evaluated two baselines to assess the effectiveness of the proposed multi-view sequence ordering method: (1) *Sort*: sequences are ordered using the proposed method; (2) *Random*: sequences are randomly ordered. The *Random* baseline led to a 42.4% (50.64%) increase in bitrate for PSNR (MS-SSIM) compared to *Sort*. Furthermore, *Sort* exhibited only a 3.76% (2.97%) bitrate increase for PSNR (MS-SSIM) compared to the manually sorted sequences in the Tanks&Temples dataset. These results demonstrate the effectiveness of the proposed ordering method for unsorted multi-view sequences, achieving performance close to that of manual sorting.

## 5 CONCLUSION

In this paper, we present 3D-GP-LMVIC, a novel learning-based multi-view image coding framework incorporating 3D Gaussian geometric priors. This framework exploits these geometric priors to estimate complex disparities and masks between views for effectively utilizing reference view information in the compression process. Additionally, we propose a depth map compression model designed to compactly and accurately represent the geometry of each view, incorporating a cross-view depth prediction module to capture inter-view geometric correlations. Moreover, we introduce a multi-view sequence ordering method for unordered sequences, enhancing the overlap between adjacent views by defining an inter-view distance measure to guide the sequence ordering. Experimental results confirm that 3D-GP-LMVIC surpasses existing learning-based coding schemes in compression efficiency while maintaining competitive coding speed.

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

APPENDIX

## A   DISPARITY AND MASK ESTIMATION ALGORITHM

---

**Algorithm 1:** Disparity and Mask Estimation

---
**Input**  : 3D-GS-based depth estimation function $GSDE$, camera intrinsic matrix $K$, camera
extrinsic matrices $V_n$ and $V_{n-1}$
**Output:** Disparity map $\boldsymbol{\Delta}_n$, mask $\boldsymbol{x}_{n,\mathrm{m}}$

1 $\boldsymbol{d}_n \leftarrow GSDE(K, V_n)$;
2 $\boldsymbol{d}_{n-1} \leftarrow GSDE(K, V_{n-1})$;
3 $\boldsymbol{\Delta}_n, \boldsymbol{d}'_{n-1} \leftarrow \text{DisparityEstimation}(\boldsymbol{d}_n, K, V_n, V_{n-1})$;
4 $\boldsymbol{x}_{n,\mathrm{m}} \leftarrow \text{MaskEstimation}(\boldsymbol{\Delta}_n, \boldsymbol{d}'_{n-1}, \boldsymbol{d}_{n-1})$;

---

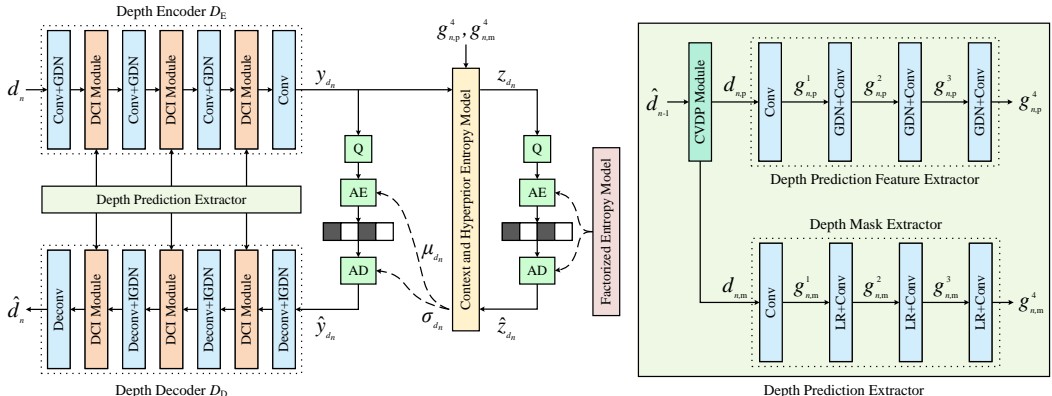

Figure 8: The architecture of the proposed depth map compression model. 'LR' represents the Leaky ReLU activation function, 'Q' denotes the quantization operation, and 'AE'/'AD' refer to the arithmetic encoder/decoder, respectively.

## B   SUPPLEMENTARY INFORMATION FOR THE DEPTH MAP COMPRESSION MODEL

As illustrated in Figure 8, during the compression and decompression of $\boldsymbol{d}_n$, $\hat{\boldsymbol{d}}_{n-1}$ is initially processed by the depth prediction extractor, which extracts multi-scale depth prediction features and associated feature masks. These extracted features and masks are then integrated into the depth backbone encoder and decoder via the depth context integration (DCI) module. Detailed explanations of the depth prediction extractor and the DCI module are provided in the subsequent content.

**Depth prediction extractor.** As illustrated in Figure 8, we first utilize the proposed cross-view depth prediction (CVDP) module to predict the depth map $\boldsymbol{d}_{n,\mathrm{p}} \in \mathbb{R}^{W \times H}$ and the associated mask $\boldsymbol{d}_{n,\mathrm{m}} \in \mathbb{R}^{W \times H}$ for the $n$-th view, based on $\hat{\boldsymbol{d}}_{n-1}$. Specifically, for each pixel $(x_{n-1}, y_{n-1})$ in the $(n-1)$-th view and its corresponding reconstructed depth $\hat{d}_{n-1}$, the CVDP module determines the corresponding pixel coordinates $(x_n, y_n)$ and the depth prediction $d'_n$ in the $n$-th view using the method described in Eq. 3. The depth at the nearest grid point $(\lfloor x_n - 0.5 \rceil, \lfloor y_n - 0.5 \rceil)$ is then set to $d'_n$:

$$\boldsymbol{d}_{n,\mathrm{p}}[\lfloor x_n - 0.5 \rceil, \lfloor y_n - 0.5 \rceil] = d'_n. \tag{10}$$

This cross-view depth prediction is applied to each pixel in the $(n-1)$-th view to construct $\boldsymbol{d}_{n,\mathrm{p}}$. If multiple pixel coordinates map to the same grid point, the depth prediction for that point is set to the minimum of these predicted depths. Additionally, if a grid point has no corresponding pixel coordinates, its depth prediction value is set to 0. The mask $\boldsymbol{d}_{n,\mathrm{m}}$ indicates whether each grid point

has at least one corresponding pixel coordinate, with values set to 1 where a correspondence exists and 0 otherwise.

Subsequently, $\boldsymbol{d}_{n,\mathrm{p}}$ is input into the depth prediction feature extractor to obtain multi-scale depth prediction features $\{\boldsymbol{g}_{n,\mathrm{p}}^i \mid i = 1, 2, 3, 4\}$. Concurrently, the mask $\boldsymbol{d}_{n,\mathrm{m}}$ is processed by the depth mask extractor to derive the associated multi-scale feature masks $\{\boldsymbol{g}_{n,\mathrm{m}}^i \mid i = 1, 2, 3, 4\}$.

**Depth Context Integration Module.** Each DCI module integrates the input features $\boldsymbol{g}_n^{i^*}$ from the backbone network with $\boldsymbol{g}_{n,\mathrm{p}}^i$ through channel-wise concatenation, followed by element-wise multiplication with $\boldsymbol{g}_{n,\mathrm{m}}^i$ to produce the output feature $\boldsymbol{g}_n^i$:

$$\boldsymbol{g}_n^i = (\boldsymbol{g}_n^{i^*} \oplus \boldsymbol{g}_{n,\mathrm{p}}^i) \odot \boldsymbol{g}_{n,\mathrm{m}}^i, \tag{11}$$

where $\oplus$ denotes channel-wise concatenation and $\odot$ denotes element-wise multiplication.

## C   PROOF OF $\mathcal{D}_{\mathcal{V}}(i, j)$ AS A DISTANCE MEASURE FOR 2-NORM AND FROBENIUS NORM

### C.1   PROOF FOR 2-NORM

#### C.1.1   DEFINITION

**Definition 1.** *For $u = (A, B)$ and $v = (C, D)$, where $A, C \in \mathbb{R}^{n \times m}$ and $B, D \in \mathbb{R}^{n \times l}$, we define $(u, v)_2 = \|AC^T + BD^T\|_2$. For any scalar $\alpha$, $\alpha u = (\alpha A, \alpha B)$. Additionally, $u + v = (A + C, B + D)$.*

#### C.1.2   LEMMA

**Lemma 1.** *For any $u = (A, B)$ and $v = (C, D)$ as defined in Definition 1, the following inequality holds:*

$$(u, v)_2 \leq \sqrt{(u, u)_2 (v, v)_2}$$

*Proof.* For any real number $t$, we have:

$$(u + tv, u + tv)_2 = \|(A + tC)(A + tC)^T + (B + tD)(B + tD)^T\|_2$$
$$\leq \|AA^T + BB^T\|_2 + t\|AC^T + BD^T\|_2 + t\|CA^T + DB^T\|_2 + t^2\|CC^T + DD^T\|_2$$
$$= (u, u)_2 + t(u, v)_2 + t(v, u)_2 + t^2(v, v)_2$$
$$= (u, u)_2 + 2t(u, v)_2 + t^2(v, v)_2$$

The right-hand side of the last equation can be viewed as a quadratic expression in $t$ and is greater than or equal to $(u + tv, u + tv)_2$, which is non-negative. Therefore, the discriminant of this quadratic must be non-positive:

$$(2(u, v)_2)^2 - 4(u, u)_2(v, v)_2 \leq 0$$

Thus, we obtain:

$$(u, v)_2 \leq \sqrt{(u, u)_2 (v, v)_2}$$

$\square$

#### C.1.3   THEOREM

**Theorem 1.** $\mathcal{D}_{\mathcal{V}}(i, j) = \|V_i V_j^{-1} - I\|_2$ *is a distance metric.*

*Proof.* We need to prove that $\mathcal{D}_{\mathcal{V}}(i, j)$ satisfies non-negativity, symmetry, and the triangle inequality.

**Non-negativity:** Since $\mathcal{D}_{\mathcal{V}}(i, j)$ is a norm, it is non-negative. Additionally, as the extrinsic matrices for different views are distinct, $V_i \neq V_j$ for $i \neq j$. $\mathcal{D}_{\mathcal{V}}(i, j) = 0$ if and only if $V_i V_j^{-1} - I = 0$, which holds only when $V_i = V_j$, i.e., $i = j$.

**Symmetry:** The extrinsic matrix $V_i$ can be represented as $V_i = \begin{pmatrix} R_i & t_i \\ 0 & 1 \end{pmatrix}$, where $R_i \in \mathbb{R}^{3\times3}$ is a rotation matrix [1] and $t_i \in \mathbb{R}^{3\times1}$ is a translation vector. We have:

$$
\begin{aligned}
\mathcal{D}_\mathcal{V}(i,j) &= \|V_i V_j^{-1} - I\|_2 \\
&= \left\| \begin{pmatrix} R_i & t_i \\ 0 & 1 \end{pmatrix} \begin{pmatrix} R_j^T & -R_j^T t_j \\ 0 & 1 \end{pmatrix} - I \right\|_2 \\
&= \left\| \begin{pmatrix} R_i R_j^T - I & -R_i R_j^T t_j + t_i \\ 0 & 0 \end{pmatrix} \right\|_2 \\
&= \left\| \begin{pmatrix} R_i R_j^T - I & -R_i R_j^T t_j + t_i \\ 0 & 0 \end{pmatrix} \begin{pmatrix} R_i R_j^T - I & -R_i R_j^T t_j + t_i \\ 0 & 0 \end{pmatrix}^T \right\|_2^{\frac{1}{2}} \\
&= \left\| 2I - R_j R_i^T - R_i R_j^T + R_i R_j^T t_j t_j^T R_j R_i^T - t_i t_j^T R_j R_i^T - R_i R_j^T t_j t_i^T + t_i t_i^T \right\|_2^{\frac{1}{2}} \\
&= \left\| R_j R_i^T (2I - R_j R_i^T - R_i R_j^T + R_i R_j^T t_j t_j^T R_j R_i^T - t_i t_j^T R_j R_i^T - R_i R_j^T t_j t_i^T + t_i t_i^T) R_i R_j^T \right\|_2^{\frac{1}{2}} \\
&= \left\| 2I - R_j R_i^T - R_i R_j^T + t_j t_j^T - R_j R_i^T t_i t_j^T - t_j t_i^T R_i R_j^T + R_j R_i^T t_i t_i^T R_i R_j^T \right\|_2^{\frac{1}{2}} \\
&= \mathcal{D}_\mathcal{V}(j,i)
\end{aligned}
$$

The fourth equality follows from the fact that for any matrix $A$, $\|A\|_2 = \|AA^T\|_2^{\frac{1}{2}}$. The sixth equality is due to the orthogonality of $R_i$ and $R_j$, and the invariance of the 2-norm under orthogonal transformations. The final equality holds because interchanging the indices $i$ and $j$ in the expression on the right-hand side of the fifth equality leads to the same expression as $\mathcal{D}_\mathcal{V}(j,i)$, which matches the right-hand side of the seventh equality.

**Triangle inequality:** For views $i$, $j$, and $k$, define $A_{i,j} = R_j^T - R_i^T$, $B_{i,j} = -R_j^T t_j + R_i^T t_i$, and similarly for $A_{j,k}, B_{j,k}, A_{k,i}, B_{k,i}$. Let $u_{j,k} = (A_{j,k}, B_{j,k})$ and $u_{k,i} = (A_{k,i}, B_{k,i})$. Starting from the fourth equation in the symmetry proof, we proceed as follows:

$$
\begin{aligned}
\mathcal{D}_\mathcal{V}(i,j) &= \left\| \begin{pmatrix} R_i R_j^T - I & -R_i R_j^T t_j + t_i \\ 0 & 0 \end{pmatrix} \begin{pmatrix} R_i R_j^T - I & -R_i R_j^T t_j + t_i \\ 0 & 0 \end{pmatrix}^T \right\|_2^{\frac{1}{2}} \\
&= \left\| (R_i R_j^T - I)(R_i R_j^T - I)^T + (-R_i R_j^T t_j + t_i)(-R_i R_j^T t_j + t_i)^T \right\|_2^{\frac{1}{2}} \\
&= \left\| R_i^T \left( (R_i R_j^T - I)(R_i R_j^T - I)^T + (-R_i R_j^T t_j + t_i)(-R_i R_j^T t_j + t_i)^T \right) R_i \right\|_2^{\frac{1}{2}} \\
&= \left\| (R_j^T - R_i^T)(R_j^T - R_i^T)^T + (-R_j^T t_j + R_i^T t_i)(-R_j^T t_j + R_i^T t_i)^T \right\|_2^{\frac{1}{2}} \\
&= \left\| A_{i,j} A_{i,j}^T + B_{i,j} B_{i,j}^T \right\|_2^{\frac{1}{2}} \\
&= \left\| (A_{j,k} + A_{k,i})(A_{j,k} + A_{k,i})^T + (B_{j,k} + B_{k,i})(B_{j,k} + B_{k,i})^T \right\|_2^{\frac{1}{2}} \\
&= \left\| A_{j,k} A_{j,k}^T + B_{j,k} B_{j,k}^T + A_{k,i} A_{k,i}^T + B_{k,i} B_{k,i}^T + A_{j,k} A_{k,i}^T + B_{j,k} B_{k,i}^T + A_{k,i} A_{j,k}^T + B_{k,i} B_{j,k}^T \right\|_2^{\frac{1}{2}} \\
&\leq \left( \|A_{j,k} A_{j,k}^T + B_{j,k} B_{j,k}^T\|_2 + \|A_{k,i} A_{k,i}^T + B_{k,i} B_{k,i}^T\|_2 + 2\|A_{j,k} A_{k,i}^T + B_{j,k} B_{k,i}^T\|_2 \right)^{\frac{1}{2}} \\
&= \left( \mathcal{D}_\mathcal{V}(j,k)^2 + \mathcal{D}_\mathcal{V}(k,i)^2 + 2(u_{j,k}, u_{k,i})_2 \right)^{\frac{1}{2}} \\
&\leq \left( \mathcal{D}_\mathcal{V}(j,k)^2 + \mathcal{D}_\mathcal{V}(k,i)^2 + 2\sqrt{(u_{j,k}, u_{j,k})_2 (u_{k,i}, u_{k,i})_2} \right)^{\frac{1}{2}} \\
&= \left( \mathcal{D}_\mathcal{V}(j,k)^2 + \mathcal{D}_\mathcal{V}(k,i)^2 + 2\mathcal{D}_\mathcal{V}(j,k)\mathcal{D}_\mathcal{V}(k,i) \right)^{\frac{1}{2}} \\
&= \mathcal{D}_\mathcal{V}(j,k) + \mathcal{D}_\mathcal{V}(k,i)
\end{aligned}
$$

---

[1] A rotation matrix is an orthogonal matrix, meaning its inverse is equal to its transpose.

The second inequality follows from Lemma 1. □

## C.2 PROOF FOR FROBENIUS NORM

### C.2.1 DEFINITION

**Definition 2.** *For $u = (A, B)$ and $v = (C, D)$ as defined in Definition 1, we define $(u, v)_F = \mathrm{tr}\left(AC^T + BD^T\right)$.*

### C.2.2 LEMMA

**Lemma 2.** *For $u$ and $v$ as defined in Definition 2, the following inequality holds:*

$$(u, v)_F \leq \sqrt{(u, u)_F (v, v)_F}.$$

*Proof.* The method of proof is analogous to that used in Lemma 1. By leveraging the properties of the trace and following a similar reasoning process, the result is derived. □

### C.2.3 THEOREM

**Theorem 2.** $\mathcal{D}_{\mathcal{V}}(i, j) = \|V_i V_j^{-1} - I\|_F$ *is a distance metric.*

*Proof.* We need to prove that $\mathcal{D}_{\mathcal{V}}(i, j)$ satisfies non-negativity, symmetry, and the triangle inequality.

**Non-negativity:** The proof follows a similar approach to that of Theorem 1, so we omit the details here.

**Symmetry:**

$$\mathcal{D}_{\mathcal{V}}(i, j) = \|V_i V_j^{-1} - I\|_F$$

$$= \left\| \begin{pmatrix} R_i & t_i \\ 0 & 1 \end{pmatrix} \begin{pmatrix} R_j^T & -R_j^T t_j \\ 0 & 1 \end{pmatrix} - I \right\|_F$$

$$= \left\| \begin{pmatrix} R_i R_j^T - I & -R_i R_j^T t_j + t_i \\ 0 & 0 \end{pmatrix} \right\|_F$$

$$= \mathrm{tr}\left( \begin{pmatrix} R_i R_j^T - I & -R_i R_j^T t_j + t_i \\ 0 & 0 \end{pmatrix} \begin{pmatrix} R_i R_j^T - I & -R_i R_j^T t_j + t_i \\ 0 & 0 \end{pmatrix}^T \right)^{\frac{1}{2}}$$

$$= \mathrm{tr}\left( 2I - R_j R_i^T - R_i R_j^T + R_i R_j^T t_j t_j^T R_j R_i^T - t_i t_j^T R_j R_i^T - R_i R_j^T t_j t_i^T + t_i t_i^T \right)^{\frac{1}{2}}$$

$$= \left( \mathrm{tr}(2I) - \mathrm{tr}(R_j R_i^T) - \mathrm{tr}(R_i R_j^T) + \mathrm{tr}(R_i R_j^T t_j t_j^T R_j R_i^T) - \mathrm{tr}(t_i t_j^T R_j R_i^T) - \mathrm{tr}(R_i R_j^T t_j t_i^T) + \mathrm{tr}(t_i t_i^T) \right)^{\frac{1}{2}}$$

$$= \left( \mathrm{tr}(2I) - \mathrm{tr}(R_j R_i^T) - \mathrm{tr}(R_i R_j^T) + \mathrm{tr}(t_j t_j^T) - \mathrm{tr}(R_i^T t_i t_j^T R_j) - \mathrm{tr}(R_j^T t_j t_i^T R_i) + \mathrm{tr}(t_i t_i^T) \right)^{\frac{1}{2}}$$

$$= \left( \mathrm{tr}(2I) - \mathrm{tr}(R_i R_j^T) - \mathrm{tr}(R_j R_i^T) + \mathrm{tr}(t_i t_i^T) - \mathrm{tr}(R_j^T t_j t_i^T R_i) - \mathrm{tr}(R_i^T t_i t_j^T R_j) + \mathrm{tr}(t_j t_j^T) \right)^{\frac{1}{2}}$$

$$= \mathcal{D}_{\mathcal{V}}(j, i)$$

The fourth equality holds because, for any matrix $A$, we have $\|A\|_F = \mathrm{tr}(AA^T)^{\frac{1}{2}}$. The sixth equality is a result of the linearity of the trace operator. The seventh equality follows from the cyclic property of the trace, for instance, $\mathrm{tr}(R_i R_j^T t_j t_j^T R_j R_i^T) = \mathrm{tr}(t_j t_j^T R_j R_i^T R_i R_j^T) = \mathrm{tr}(t_j t_j^T)$.

**Triangle Inequality:** For views $i$, $j$, and $k$, we follow the same definitions of $A_{i,j}$, $B_{i,j}$, $A_{j,k}$, $B_{j,k}$, $A_{k,i}$, $B_{k,i}$, $u_{j,k}$, and $u_{k,i}$ as in the proof of the triangle inequality in Theorem 1. Starting from the fourth equality in the proof of symmetry, we have:

$$\mathcal{D}_{\mathcal{V}}(i, j) = \mathrm{tr}\left( \begin{pmatrix} R_i R_j^T - I & -R_i R_j^T t_j + t_i \\ 0 & 0 \end{pmatrix} \begin{pmatrix} R_i R_j^T - I & -R_i R_j^T t_j + t_i \\ 0 & 0 \end{pmatrix}^T \right)^{\frac{1}{2}}$$

$$= \mathrm{tr}\left( (R_i R_j^T - I)(R_i R_j^T - I)^T + (-R_i R_j^T t_j + t_i)(-R_i R_j^T t_j + t_i)^T \right)^{\frac{1}{2}}$$

$$= \text{tr} \left( R_i^T \left( (R_i R_j^T - I)(R_i R_j^T - I)^T + (-R_i R_j^T t_j + t_i)(-R_i R_j^T t_j + t_i)^T \right) R_i \right)^{\frac{1}{2}}$$

$$= \text{tr} \left( (R_j^T - R_i^T)(R_j^T - R_i^T)^T + (-R_j^T t_j + R_i^T t_i)(-R_j^T t_j + R_i^T t_i)^T \right)^{\frac{1}{2}}$$

$$= \text{tr} \left( A_{i,j} A_{i,j}^T + B_{i,j} B_{i,j}^T \right)^{\frac{1}{2}}$$

$$= \text{tr} \left( (A_{j,k} + A_{k,i})(A_{j,k} + A_{k,i})^T + (B_{j,k} + B_{k,i})(B_{j,k} + B_{k,i})^T \right)^{\frac{1}{2}}$$

$$= \text{tr} \left( A_{j,k} A_{j,k}^T + B_{j,k} B_{j,k}^T + A_{k,i} A_{k,i}^T + B_{k,i} B_{k,i}^T + A_{j,k} A_{k,i}^T + B_{j,k} B_{k,i}^T + A_{k,i} A_{j,k}^T + B_{k,i} B_{j,k}^T \right)^{\frac{1}{2}}$$

$$= \left( \text{tr} \left( A_{j,k} A_{j,k}^T + B_{j,k} B_{j,k}^T \right) + \text{tr} \left( A_{k,i} A_{k,i}^T + B_{k,i} B_{k,i}^T \right) + 2\text{tr} \left( A_{j,k} A_{k,i}^T + B_{j,k} B_{k,i}^T \right) \right)^{\frac{1}{2}}$$

$$= \left( \mathcal{D}_{\mathcal{V}}(j,k)^2 + \mathcal{D}_{\mathcal{V}}(k,i)^2 + 2(u_{j,k}, u_{k,i})_F \right)^{\frac{1}{2}}$$

$$\leq \left( \mathcal{D}_{\mathcal{V}}(j,k)^2 + \mathcal{D}_{\mathcal{V}}(k,i)^2 + 2\sqrt{(u_{j,k}, u_{j,k})_F (u_{k,i}, u_{k,i})_F} \right)^{\frac{1}{2}}$$

$$= \left( \mathcal{D}_{\mathcal{V}}(j,k)^2 + \mathcal{D}_{\mathcal{V}}(k,i)^2 + 2\mathcal{D}_{\mathcal{V}}(j,k)\mathcal{D}_{\mathcal{V}}(k,i) \right)^{\frac{1}{2}}$$

$$= \mathcal{D}_{\mathcal{V}}(j,k) + \mathcal{D}_{\mathcal{V}}(k,i)$$

The third equality holds because the trace is invariant under similarity transformations. □

## D EXPERIMENTAL DETAILS

**Datasets.** Our evaluation is conducted on three multi-view image datasets: Tanks&Temples, Mip-NeRF 360, and Deep Blending. Tanks&Temples consists of 21 diverse indoor and outdoor scenes, ranging from sculptures and large vehicles to complex large-scale environments, with intricate geometry and varied lighting conditions. Mip-NeRF 360 includes 9 scenes—5 outdoor and 4 indoor—captured in unbounded settings, allowing for 360-degree camera rotations and capturing content at varying distances. From the Deep Blending dataset, we selected 9 representative scenes that span indoor, outdoor, vegetation-rich, and nighttime environments. For all datasets, 90% of the images in each scene were allocated for training, with the remaining 10% used for testing.

**Benchmarks.** We assess the coding performance of MV-HEVC using the HTM-16.3 software[2]. The learning-based multi-view image codecs used as baselines, along with our proposed method, are trained under the same conditions on a shared training set and evaluated on a common test set. For the 3D Gaussian Splatting compression method (HAC), we train the 3D Gaussian representations on each scene's test data and measure the reconstruction quality of the rendered images. The bpp is determined by dividing the size of the compressed 3D Gaussian file by the total number of pixels in the test images.

**Implementation Details.** We utilize the Adam optimizer for training with a batch size of 2. To facilitate data augmentation and optimize memory usage, each image is randomly cropped to $256 \times 256$. Correspondingly, the principal point in the intrinsic matrix $K$ is adjusted to reflect the new crop. The intrinsic matrix $K$ is given by:

$$K = \begin{pmatrix} f_x & 0 & c_x \\ 0 & f_y & c_y \\ 0 & 0 & 1 \end{pmatrix},$$

where $f_x$ and $f_y$ represent the focal lengths along the x and y axes, respectively, and $c_x$ and $c_y$ are the principal point coordinates. If the top-left corner of the crop is located at $(p_x, p_y)$ in the original image, the updated intrinsic matrix $K'$ becomes:

$$K' = \begin{pmatrix} f_x & 0 & c_x - p_x \\ 0 & f_y & c_y - p_y \\ 0 & 0 & 1 \end{pmatrix}.$$

---

[2]https://vcgit.hhi.fraunhofer.de/jvet/HTM/-/tags

Table 3: Average alignment quality (PSNR, MS-SSIM) and runtime of different alignment methods on the Train scene of the Tanks&Temples dataset.

| Metrics | Methods | | | | | | |
|---------|------|------|-------|---------|-------------|------------------|--------|
|         | HT   | PM   | SPyNet | PWC-Net | FlowFormer++ | 3D-GP-A (original) | 3D-GP-A |
| PSNR    | 15.16 | 17.94 | 16.12 | 17.59 | 18.08 | 17.36 | 18.14 |
| MS-SSIM | 0.5435 | 0.7633 | 0.6289 | 0.7707 | 0.7863 | 0.7410 | 0.8053 |
| Runtime | 43ms | 207ms | 7ms | 11ms | 836ms | 26ms | 14ms |

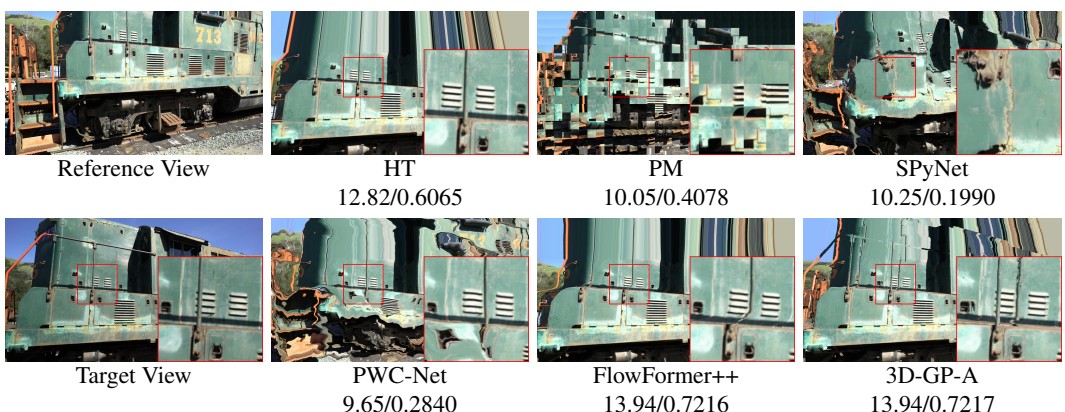

Figure 9: Visual comparison of different alignment methods on an adjacent view pair in the Train scene of the Tanks&Temples dataset. Alignment quality is reported as PSNR/MS-SSIM.

**Ablation study details.** To implement *Separate*, we set the reference view images, predicted depth maps, and masks to full-zero tensors, with $\lambda_{\text{dep}}$ set to zero. In *Concatenation*, alignment operations in the ICT modules are removed. For *W/O Mask*, we eliminate all mask-related multiplications in the ICT and DCI modules. In *W/O Dep.Pred*, the predicted depth maps are replaced with full-zero tensors. For both *Sort* and *Random*, sequences in the training and test sets are reordered accordingly.

# E  ALIGNMENT EXPERIMENTS

To evaluate the effectiveness of the proposed 3D Gaussian geometric priors-based alignment method (3D-GP-A), we conducted alignment experiments on the Train scene from the Tanks&Temples dataset. The baselines for comparison include alignment methods commonly used in learning-based multi-view image codecs, such as homography transformation (HT) (Deng et al., 2021) and patch matching (PM) (Huang et al., 2023), optical flow estimation methods, including SPyNet (Ranjan & Black, 2017), PWC-Net (Sun et al., 2018), and FlowFormer++ (Shi et al., 2023), as well as a depth map estimation method based on the original 3D-GS (Kerbl et al., 2023). Alignment quality was assessed by computing PSNR and MS-SSIM between the aligned reference view images and the target view images. Table 3 summarizes the average alignment quality and runtime for each method. The proposed 3D-GP-A method outperformed the baselines in both PSNR and MS-SSIM, indicating its effectiveness in capturing complex disparities between views while maintaining competitive runtime performance. Figure 9 provides visual comparisons, demonstrating that 3D-GP-A achieves closer alignment with the target view images. However, in the magnified red box, two iron bars appear, while only the left bar exists in the target view. This ghosting effect occurs because the train's outer shell behind the right bar in the target view is occluded in the reference view, causing misalignment. The mask $x_{n,\text{m}}$ in Eq. 4 can indicate occluded regions. Figure 10 shows visual examples of 3D-GP-A along with the corresponding masks. Notably, ghosting artifacts due to occlusion, such as those involving the iron bars and the edge of the train shell, are effectively identified by the mask, aiding the codec in filtering out irrelevant information when merging features from the reference view.

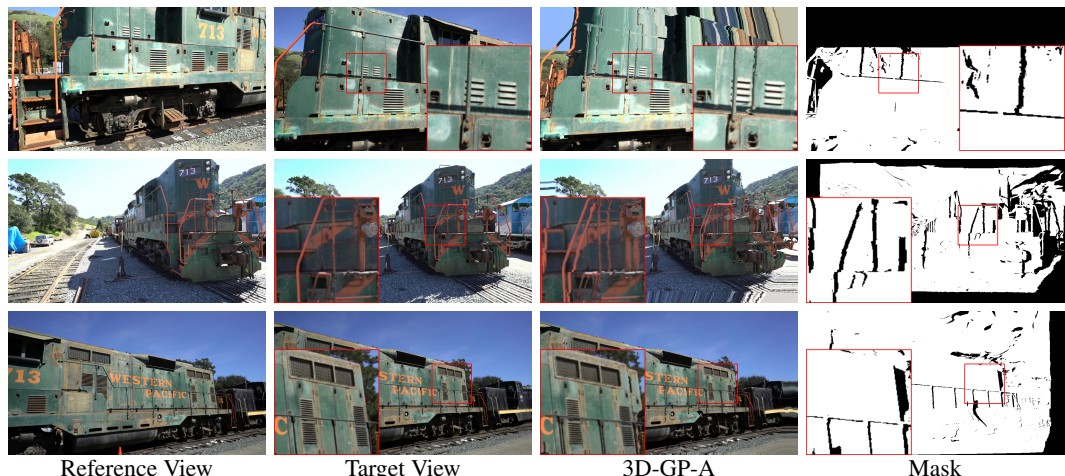

| Reference View | Target View | 3D-GP-A | Mask |

Figure 10: Visual examples of 3D-GP-A and the mask from Eq. 4.

Table 4: BDBR of 3D-GP-LMVIC relative to HEVC.

| Methods | Tanks&Temples | | Mip-NeRF 360 | | Deep Blending | |
|---------|------|---------|------|---------|------|---------|
| | PSNR | MS-SSIM | PSNR | MS-SSIM | PSNR | MS-SSIM |
| 3D-GP-LMVIC | -20.69% | -40.75% | -14.48% | -22.06% | -17.29% | -43.06% |

## F SUPPLEMENTARY CODING PERFORMANCE

We present a supplementary comparison of the coding performance between the proposed 3D-GP-LMVIC and the HEVC video coding standard. The multi-view sequences are treated as a single video and compressed using HEVC with the *lowdelay_P* configuration and YUV444 input format. HEVC's coding efficiency is evaluated using the HM-18.0 software[3]. Table 4 reports the BDBR of 3D-GP-LMVIC relative to HEVC. On the three datasets, 3D-GP-LMVIC consistently surpasses HEVC in both PSNR and MS-SSIM, demonstrating its effectiveness in reducing inter-view redundancy in multi-view sequences.

---

[3]https://vcgit.hhi.fraunhofer.de/jvet/HM/-/tags

