# OpenReview forum: "3D-GP-LMVIC: Learning-based Multi-View Image Compression with 3D Gaussian Geometric Priors"
_ICLR.cc/2025/Conference — Submitted to ICLR 2025_

### Official Review · Reviewer_c6CX · 2024-10-29

**Soundness:** 3
**Presentation:** 2
**Contribution:** 2
**Rating:** 5
**Confidence:** 4

**Summary:**

This work presents a new method for multi-view image compression (MVIC) based on deep neural networks. Considering that accurate disparity estimation is a key factor in MVIC, authors proposed to leverage 3D Gaussian splatting for better disparity estimation. Then, the images and estimated depth maps are compressed in an encoder-decoder framework, by making use of mask maps. The multi-view sequence ordering is also proposed using a simple greedy method to maximize the inter-view correlation. Experiments demonstrate the effectiveness of the proposed codec over existing methods.

**Strengths:**

1) Experiments demonstrates the effectiveness of the proposed method over existing MVIC approaches.

**Weaknesses:**

1) Contributions should be clarified. It seems that the depth estimation method itself of Section 3.1 is not new, following existing approaches. To be specific, the original 3D Gaussian splitting method [a] can also extract depth maps from the rendering process, and recently some approaches attempt to enhance the depth quality from the 3D Gaussian splitting framework [b]. If there is nothing new in terms of the depth estimation process, this part can be excluded from Section 3 (Proposed Method).

[a] 3D Gaussian Splatting for Real-Time Radiance Field Rendering, ACM TOG 2023

[b] Self-Evolving Depth-Supervised 3D Gaussian Splatting from Rendered Stereo Pairs, BMVC 2024

2) Section 3.2 about image and depth compression needs significant revisions.
- The compression methodologies of (5) and (6) consist of encoder, quantization, and decoder. It is difficult to grab what the differences from existing learning based MVIC approaches are. More concrete explanations against existing MVIC methods (cited in the paper) should be included.

- What is the purpose of using the mask map in the encoding process?

- Figure 3 is complicate, and it would be better to visualize it with high-level conceptual figure, followed by detailed architectures of separate modules.

- $y_n$ is quantized in (4) and (5), but $z_n$ is quantized in Figure 3.

- Why is 'MaskConv' used in Figure 3?

3) Minor comments
(3) represents the equation of converting a depth into a disparity, and is commonly used in many literatures. So, it would be better to move into appendix instead of main paper.

Overall, this work achieves satisfactory results compared to recent learning-based MVIC, but the technical presentation needs substantial revisions.

**Questions:**

Refer to the comments in the weaknesses.

**Details Of Ethics Concerns:**

N.A.

---

> ### Author Response · Authors · 2024-11-22
>
> hanks to you for the valuable comments. We are appreciated for recognizing the strengths of our paper in terms of performance. We will alleviate your remaining concerns as follows:
>
> ### R1[The novelty of the depth estimation process]
> ---
> Our depth map estimation based on the 3D Gaussian representation differs from the original 3D Gaussian Splatting method [1] and is not mentioned in [2]. This difference improves disparity estimation accuracy compared to the original method, which calculates pixel depth using a weighted averaging approach:
> $$
> d = \sum_{i=1}^{M} T_i\alpha_i z_i,
> $$
> where $T_i, \alpha_i, z_i$ represent the transmittance, opacity, and depth of the i-th point, respectively. In our work, we adopt a median-based estimation approach:
> $$
> d = z_{i^*}, \hspace{0.5em} \text{where} \hspace{0.5em} i^* = \min ( i \mid T_i < 0.5 ).
> $$
> Since the transmittance decreases as light passes through the points, starting from an initial value of 1, we take the depth of the point at which the transmittance drops below 0.5 as the estimated depth value.
>
> We tested the alignment performance of the original 3D Gaussian Splatting depth estimation method under the same experimental settings as described in Appendix E of this paper. Since the code for [2] does not appear to be publicly available, we were unable to test the method from [2]. We combined the results with Table 3, as shown in the table below:
> | Metrics | HT | PM | SPyNet| PWC-Net | FlowFormer++ | 3D-GP-A (original) | 3D-GP-A |
> |:-------:|:-------:|:-------:|:-------:|:-------:|:-------:|:-------:|:-------:|
> | PSNR↑ | 15.16 | 17.94 | 16.12 | 17.59 | 18.08 | 17.36 | 18.14 |
> | MS-SSIM↑ | 0.5435 | 0.7633 | 0.6289 | 0.7707 | 0.7863 | 0.7410 | 0.8053 |
>
> **3D-GP-A outperforms 3D-GP-A (original) in PSNR and MS-SSIM**, indicating that the proposed depth estimation method improves the alignment accuracy compared to the original depth estimation method used in 3D Gaussian Splatting.
>
> **We have explained the differences between our depth map estimation method and the original 3D-GS depth estimation method in lines 204–208 of the main text. Additionally, we have included a comparison of alignment performance with the original 3D-GS depth estimation method in Appendix E.**
>
> [1] Kerbl B, Kopanas G, Leimkühler T, et al. 3D Gaussian Splatting for Real-Time Radiance Field Rendering. ACM TOG, 2023.
>
> [2] Safadoust S, Tosi F, Güney F, et al. Self-Evolving Depth-Supervised 3D Gaussian Splatting from Rendered Stereo Pairs. BMVC, 2024.
>
> ### R2[Section 3.2 needs revision]
> ---
> Thank you for your suggestion. **We have made significant revisions to Section 3.2, which you can find in the updated paper. Alternatively, you may refer to the following response.**
>
> ### R2.1[Explanation of differences compared to existing MVIC methods in (5) and (6)]
> ---
> Thank you for your suggestion. We have revised the descriptions in **lines 268–289 (5)** and **lines 320–328 (7) (previously (6))** to highlight the unique aspects of our compression model compared to other methods.
>
> In (5), we primarily designed the disparity extractor and reference feature extractor. These two modules help the compression model integrate aligned reference features into the main network to improve compression efficiency.
>
> In (6), we introduced the depth prediction extractor. This module predicts the current view's depth map based on the reference view's depth map, thereby reducing geometric redundancy between views.
>
> ### R2.2[Purpose of the mask map in encoding]
> ---
> **The mask map is designed to identify non-overlapping regions between viewpoints and mask out irrelevant information in the reference features, thereby avoiding the introduction of noise.** When capturing a scene from different viewpoints, an object may appear in one view but not in another. The mask map's purpose is to determine whether a pixel in the current view corresponds to the same object as the aligned pixel in the reference view. **We have added the motivation for incorporating the mask map in lines 315–316.**
>
> The ablation study in Section 4.3 compares the proposed method with *W/O Mask*, demonstrating that incorporating the mask map contributes to improved compression efficiency.
>
> ### R2.3[Figure 3 is complicate]
> ---
> Thank you for your suggestion. **We have simplified Figure 3 by dividing the flowchart into a high-level overview of the overall process and detailed components for each module.** You can find the corresponding changes in the revised paper.
>
> ### R2.4[$y_n$ is quantized in (4) and (5), but $z_n$ is quantized in Figure 3]
> ---
> Thank you for your suggestion. We have added descriptions in lines 293–296 and 331–333 of the paper, explaining the quantization of \(z_n\) and \(z_{d_n}\) and the modeling of the latent variable probability distribution based on the quantized hyperprior representation. Since the entropy model primarily uses existing methods, we have kept this part concise.

---

> ### Author Response · Authors · 2024-11-25
>
> ### R2.5[Why is 'MaskConv' used in Figure 3?]
> ---
> In the autoregressive entropy model, 'MaskConv' is used to divide the latent variable \(y_n\) into disjoint segments, which are then encoded and decoded sequentially. Following your suggestion in 'R2.3', we have simplified Figure 3 by representing the context and hyperprior entropy model, which are not part of our proposed method, with a single block. As a result, 'MaskConv' has also been omitted. This revision makes the flowchart more concise and highlights our proposed approach.
>
> ### R3[Suggestion to move (3) into appendix]
> ---
> Thank you for your suggestion. Since the notation $d'_{n-1}$ in (3) is used in (4), and Section 3.3 defines the inter-view distance based on (3), we have decided to retain (3) in the main text for now. This ensures continuity in the article and improves readability for the audience.

---

> ### Author Response · Authors · 2024-11-25
>
> Thank you once again for dedicating your valuable time to reviewing our paper and providing constructive comments!
>
> As the end of the discussion period approaches, we kindly ask if our responses, which address all the questions and concerns you raised, have satisfactorily resolved your queries. Your feedback would be greatly appreciated, and we would be delighted to engage in further discussions if needed.
>
> Sincerely,
> The Authors

---

> ### Author Response · Authors · 2024-11-27
> **To Reviewer c6CX:**
>
> Thank you again for reviewing our paper. We have addressed all the issues you raised and believe our responses sufficiently resolve your concerns. We have revised the description and flowchart in Section 3.2 to highlight the proposed components of the image compression model, including the disparity extractor, reference feature extractor, and image context transfer module. Additionally, we clarified the differences between our depth estimation method and the original approach used in 3D-GS, as well as the improvements in alignment performance resulting from these differences.
>
> We hope you will recognize the revisions of our paper and consider revising your score. Should you have any further questions, we would be happy to provide additional clarifications.

---

> ### Author Response · Authors · 2024-11-28
> **A Second Reminder of the Post-rebuttal Feedback**
>
> We deeply value your initial feedback and understand that your schedule may be demanding. However, we kindly request that you take a moment to review our responses to your concerns.
>
> We have made substantial revisions to Section 3.2, including simplifying the parts of the flowchart that describe prior work and emphasizing the details of the proposed method. Equations (5) and (7) now highlight the unique contributions of our disparity extractor and depth prediction extractor. Additionally, we have focused the writing more explicitly on our contributions. All your suggestions regarding Section 3.2 have been considered, and the necessary modifications have been made.
>
> For Section 3.1, we have added clarifications to distinguish our depth estimation method from the one used in the original 3D-GS framework. Furthermore, we demonstrated through experiments that our proposed method achieves superior alignment performance.
>
> Lastly, regarding your suggestion to move Equation (3) to the appendix, we considered its critical role in defining the notation used in Equation (4) and in Section 3.3, where we propose the view-to-view distance metric based on insights derived from Equation (3). To maintain coherence and readability, we have retained Equation (3) in the main text for now.
>
> In conclusion, we kindly ask for your response once more, as we are eager to engage in further discussion with you.

---

> ### Author Response · Authors · 2024-11-29
> **Our Contributions and Significance: We Look Forward to Your Feedback**
>
> **Thank you again for reviewing our paper!** In addition to addressing your comments on the writing, we would like to further elaborate on the motivation, contributions, and significance of our work for your consideration.
>
> ---
>
> ### **1. Motivation and Practical Applications**
> Our goal is to design a multi-view image compression framework tailored for wide-baseline setups to achieve accurate disparity estimation and effectively eliminate inter-view redundancy. Unlike prior works that focus on compressing stereo images—captured by closely positioned cameras and characterized by small, primarily horizontal disparities—we address wide-baseline setups, where data features irregular view relationships and less consistent disparities. These challenges render existing disparity estimation methods, such as homography transformation and patch matching, less effective, as demonstrated in Table 3 on the TnT dataset.
>
> Wide-baseline setups are crucial for practical applications, where scenes often consist of dozens to hundreds of images, posing significant challenges for storage and transmission.
>
> ---
>
> ### **2. Contributions**
> - We propose leveraging **3D-GS to estimate depth maps**, enabling accurate disparity estimation under wide-baseline setups.
> - In the image compression model, we introduce a **context transfer module** for aligning features across viewpoints while masking non-overlapping regions between views.
> - In the depth compression model, we design a **depth prediction extractor** to predict the depth map of the current view based on the reference view’s depth map, effectively reducing geometric redundancy between views.
> - We develop a **multi-view sequence sorting method** to maximize similarity between adjacent views, improving compression efficiency.
>
> We validated these contributions through comprehensive alignment and ablation experiments.
>
> ---
>
> ### **3. Significance**
> We believe our work offers a novel perspective in the multi-view image domain by shifting the focus from sparse-view image compression (e.g., stereo images) to wide-baseline setups. The latter captures more comprehensive scene information from diverse viewpoints, which is becoming increasingly relevant as 3D applications continue to grow. This type of data is inherently large-scale, making efficient compression crucial.
>
> Additionally, we implemented a complete encoding and decoding pipeline and will release the code to enable further research and practical applications in this area.
>
> ---
>
> We respect your evaluation and sincerely hope you recognize the significance and value of our work. We look forward to hearing your response!

---

> ### Author Response · Authors · 2024-12-01
> **Follow-Up on Rebuttal Response to Reviewer c6CX**
>
> Dear Reviewer c6CX,
>
> We greatly appreciate the time and effort you have already dedicated to reviewing our work. As the discussion phase is nearing its conclusion, we kindly request your feedback regarding our responses to your comments.
>
> To summarize, here are the key updates and clarifications we have made in response to your review:
>
> 1. **Section 3.2 Improvements**:
>    - We revised the writing and reorganized the flowchart to simplify the parts borrowed from existing works while emphasizing the details of our proposed methods, including the *disparity extractor*, *reference feature extractor*, and *image context transfer module*.
>    - Equations (5) and (7) now clearly highlight the unique aspects of our approach.
>
> 2. **Clarification of Depth Estimation**:
>    - We explicitly explained the differences between our depth estimation method and the original 3D-GS framework, highlighting the resulting improvements in alignment performance.
>
> 3. **Response to Your Suggestions**:
>    - Regarding your suggestion to move Equation (3) to the appendix, we provided a rationale for keeping it in the main text due to its relevance to Equation (4) and its role in motivating the view-sorting mechanism described in Section 3.3.
>
> We believe these revisions address your concerns and enhance the clarity and strength of the paper. Your feedback on these updates is invaluable to us, and we hope for your response to ensure a productive discussion.
>
> Thank you again for your time and support.
>
> Best regards,
> The Authors of "3D-GP-LMVIC: Learning-based Multi-View Image Compression with 3D Gaussian Geometric Priors"

---

> ### Author Response · Authors · 2024-12-03
> **Friendly Reminder Regarding Rebuttal Discussion**
>
> Dear Reviewer c6CX,
>
> As the discussion phase is coming to a close, we would like to kindly remind you of our responses to your valuable comments and suggestions. We have thoroughly addressed the concerns you raised, including:
>
> 1. Revising Section 3.2 to improve the description and flowchart, with a focus on highlighting the unique contributions of our method.
> 2. Clarifying the differences between our depth estimation method and the original approach used in 3D-GS, demonstrating the resulting improvements in alignment performance through experiments.
>
> We deeply value your feedback and hope you will have the opportunity to share any further thoughts before the discussion period concludes. Your insights are greatly appreciated and will help us further enhance the quality of our work.
>
> Thank you once again for your time and effort in reviewing our paper.
>
> Best regards,
> The Authors of "3D-GP-LMVIC: Learning-based Multi-View Image Compression with 3D Gaussian Geometric Priors"

---

### Official Review · Reviewer_MS7A · 2024-11-03

**Soundness:** 3
**Presentation:** 3
**Contribution:** 2
**Rating:** 6
**Confidence:** 4

**Summary:**

This paper introduces 3D-GP-LMVIC, a novel method for multi-view image compression that uses 3D Gaussian Splatting to improve disparity estimation in complex, wide-baseline scenarios. It includes a depth map compression model to minimize geometric redundancy and a multi-view sequence ordering method to enhance view correlations. Experiments show that 3D-GP-LMVIC outperforms existing methods in compression efficiency while keeping fast processing speeds.

**Strengths:**

(1) In this paper, the authors carry out a lot of formula derivation to explain 3D Gaussian geometric priors.

(2) 3D-GP-LMVIC achieves SOTA performance on Tanks&Temples, Mip-NeRF 360, Deep Blending dataset compared with other deep learning-based multi-view image compression methods.

**Weaknesses:**

(1) Although the author used a large number of formulas to derive 3DGS prior in this paper, I think the interpretation of 3DGS prior is not clear enough. Could you elaborate on how the 3D Gaussian Splatting technique is used to derive geometric priors? How do these priors differ from traditional disparity information, and what unique advantages do they offer for multi-view image compression?

(2) In the selection of datasets in the experimental part, the authors select several datasets commonly used by 3DGS methods. I'm curious why the author didn't add the commonly-used multi-view image compression dataset Cityspace? In addition, since the author emphasizes in the abstract that the application scenarios of the existing methods are mainly stereo images, I strongly suggest that the author include the performance of 3D-GP-LMVIC on stereo image datasets (such as KITTI and Instereo2K).

(3) Are there depth map instances in the 3 datasets used by the authors? If so, why did the authors not show RD performance for depth map compression? If not, I observe that 3D-GP-LMVIC seems to be a dual framework. Did the author include the depth map when calculating Bpp (bits per pixel)? Is this cost-effective relative to the performance improvement, and is there any corresponding experimental proof?

(4) I'm curious why the authors do not show the BiSIC codec time in Table 2. In addition, I suggest that the authors further supplement the number of model parameters for each method to evaluate the spatial complexity of each model.

If the authors can solve the above problems I raised in the discussion stage, I am willing to raise my score.

**Questions:**

Please see Weaknesses.

---

> ### Author Response · Authors · 2024-11-22
>
> Thanks to you for the valuable comments. We are appreciated for recognizing the strengths of our paper in terms of performance. We will alleviate your remaining concerns as follows:
>
> ### R1[How 3D Gaussian Splatting derives geometric priors and its advantages]
> ---
> **1. 3D Gaussian Representation and Geometry Modeling**
> 3D Gaussian Splatting uses a large number of views to train a dense 3D Gaussian representation that captures the entire scene’s geometry. This allows for novel view synthesis by ensuring the representation contains sufficient geometric information. Without accurate geometric priors, rendering from arbitrary viewpoints may produce inconsistencies.
>
> **2. Depth Estimation with 3D Gaussian Representation**
> Depth estimation involves tracing a ray from the camera center through a pixel and identifying all intersecting 3D Gaussians. The depth is determined using a transmittance-based approach: the first 3D Gaussian reducing transmittance below 0.5 is taken as the pixel’s depth. This method accounts for occlusions and ensures depth accuracy.
>
> **3. Advantages Over Traditional Methods**
> Traditional disparity estimation relies on local similarity between stereo images, which works well for small viewpoint changes. However, in wide-baseline setups with significant geometric differences, such methods often fail. In contrast, the proposed 3D Gaussian-based disparity estimation method align 3D points across views, handling large viewpoint variations effectively. **Since 3D Gaussian Splatting models the entire scene using a large number of views, it comprehensively captures 3D spatial relationships and provides accurate geometric information for disparity estimation.** Appendix E of this paper compares the alignment performance of traditional disparity estimation methods and the proposed method for wide-baseline multi-view data. Experimental results demonstrate the superiority of the proposed method.
>
> **4. Benefits for Compression Efficiency**
> Accurate disparity estimation using 3D Gaussians enables precise alignment of features across views, effectively eliminating redundant information. This significantly improves compression efficiency for wide-baseline multi-view datasets.
>
> ### R2[Why use Mip-NeRF 360 and TnT datasets instead of stereo image datasets]
> ---
> **1. Motivation and Practical Applications**
> As mentioned above, we believe that previous multi-view image compression methods have already achieved satisfactory disparity estimation on stereo image datasets. However, their performance on wide-baseline setups for multi-view datasets is still insufficient. Therefore, we mainly focus on designing a multi-view image compression framework for wide-baseline setups to achieve more accurate disparity estimation and effectively exploit this estimation to eliminate redundancy across views. **Additionally, we believe that multi-view image compression under wide-baseline setups is essential for practical applications, as a single scene in such setups often consists of dozens to hundreds of images, posing significant challenges for data storage and transmission.**
>
> **2. Limitations of Stereo Images for Learning Geometry in 3D Gaussian Representation**
> We found that for stereo image datasets with only two viewpoints, the 3D Gaussian representation struggles to learn accurate geometric information. **In contrast, multi-view data under wide-baseline setups provides rich scene information sampled from widely varying viewpoints, which helps the 3D Gaussian representation learn accurate geometry.** In stereo image datasets, the cameras are often positioned close to each other with similar orientations, making it difficult for the 3D Gaussian representation to correct geometric errors through large viewpoint variations and learn the correct geometric structure. The table below presents the alignment performance of various disparity estimation methods on the Cityscapes and KITTI Stereo datasets under the same experimental settings as the alignment experiments described in Appendix E of this paper.
> | Datasets | Metrics | HT | PM | SPyNet| PWC-Net | FlowFormer++ | 3D-GP-A |
> |:-------:|:-------:|:-------:|:-------:|:-------:|:-------:|:-------:|:-------:|
> | Cityscapes | PSNR↑ | 24.43 | 27.40 | 28.63 | 29.16 | 27.26 | 14.64 |
> |  | MS-SSIM↑ | 0.7906 | 0.9546 | 0.9598 | 0.9616 | 0.8864 | 0.3934 |
> | KITTI Stereo | PSNR↑ | 14.05 | 18.33 | 18.11 | 18.92 | - | 7.77 |
> |  | MS-SSIM↑ | 0.5855 | 0.8691 | 0.8768 | 0.8952 | - | 0.1512 |
>
> Experiments have shown that the 3D Gaussian representation cannot learn accurate geometry based on stereo image data with only two viewpoints. Additionally, the disparity estimation modules in current multi-view image codecs, such as Patch Matching (PM), already perform well on stereo image datasets. Therefore, we did not conduct further tests on stereo image datasets.

---

> ### Author Response · Authors · 2024-11-22
>
> ### R3[Depth map bpp calculation and effectiveness]
> ---
> We calculated the bpp of the depth map in the proposed 3D-GP-LMVIC framework. The effectiveness of the depth map is evaluated in the ablation study in Section 4.3 of the paper. In the ablation study, *Concatenation* does not calculate the bpp of the depth map and does not use depth map-based disparity estimation to align features between two views. Instead, it directly concatenates the features of the reference view and the target view. **The rate-distortion curves in Figure 5 show that 3D-GP-LMVIC improves compression efficiency by approximately 33% compared to *Concatenation*.** This demonstrates that incorporating the depth map is beneficial.
>
> ### R4[BiSIC codec time and computational complexity analysis]
> ---
> Thank you for your suggestion. The open-source code of BiSIC does not provide separate implementations for encoding, decoding, or entropy coding, so we only measured its overall computational complexity. Since BiSIC is an improvement over LDMIC with a more complex autoencoder structure and entropy model, we infer that BiSIC's encoding and decoding time is longer than that of LDMIC, and BiSIC-Fast's encoding and decoding time is longer than that of LDMIC-Fast.
>
> Additionally, **we evaluated the Multiply-Accumulate Operations (MACs), model parameters, coding speed, and memory usage of various learning-based image codecs under the same experimental environment and settings as described in Section 4.2 of this paper for measuring encoding and decoding speed.** Furthermore, we tested a lightweight version of 3D-GP-LMVIC, where the feature channel dimensions of the autoencoder and entropy model were reduced from (128, 192) to (100, 128). The experimental results are shown in the table below.
> | Methods | MACs(Enc) | MACs(Dec) | Params(Enc) | Params(Dec) | Runtime(Enc) | Runtime(Dec) | Memory usage |
> |:-------:|:-------:|:-------:|:-------:|:-------:|:-------:|:-------:|:-------:|
> | HESIC+ | 48.16G | 134.31G | 17.18M | 15.1M | 4.35s | 10.73s | 2248M |
> | MASIC | 65.62G | 511.34G | 32.03M | 30.73M | 4.38s | 10.78s | 5202M |
> | SASIC | 91.80G | 438.09G | 3.57M | 4.44M | 0.06s | 0.09s | 4498M |
> | LDMIC-Fast | 37.49G | 94.43G | 7.73M | 11.15M | 0.11s | 0.09s | 1168M |
> | LDMIC | 30.91G | 87.84G | 7.73M | 11.15M | 4.24s | 10.63s | 1096M |
> | BiSIC-Fast | 1880G (Enc+Dec) |  | 85.9M (Enc+Dec) |  | - | - | 3552M |
> | BiSIC | 1770G (Enc+Dec) |  | 78.21M (Enc+Dec) |  | - | - | 3006M |
> | 3D-GP-LMVIC (100, 128) | 280.41G | 254.26G | 24.75M | 24.75M | 0.16s | 0.16s | 1532M |
> | 3D-GP-LMVIC (128, 192) | 479.43G | 436.16G | 41.92M | 36.87M | 0.19s | 0.18s | 3164M |
>
> The MACs of the encoder in our method are relatively large, but they should still be smaller than those of BiSIC. Since BiSIC uses a symmetric encoder-decoder structure, we estimate that the MACs for its encoder and decoder each account for approximately half of the total MACs. Furthermore, in addition to designing an image codec, we also designed a depth map codec, which is another reason for the larger MACs and model parameter count. However, overall, the MACs of 3D-GP-LMVIC are still lower than those of the current SOTA scheme, BiSIC. **The lightweight version of 3D-GP-LMVIC ranks in the middle among the compared methods in terms of MACs and model parameters, while its memory usage is quite low at only 1532M, similar to LDMIC-Fast.**
>
> The table below shows the BDBR comparison of the lightweight 3D-GP-LMVIC and other methods relative to MV-HEVC on the DeepBlending dataset.
>
> | BDBR | HESIC+ | MASIC | SASIC| LDMIC-Fast | LDMIC | BiSIC-Fast | BiSIC | 3D-GP-LMVIC (100, 128) | 3D-GP-LMVIC (128, 192) |
> |:-------:|:-------:|:-------:|:-------:|:-------:|:-------:|:-------:|:-------:|:-------:|:-------:|
> | PSNR↓ | 32.5% | 43.6% | 24.64% | 24.25% | 16.88% | -8.24% | -15.46% | -24.15% | -27.31% |
> | MS-SSIM↓ | -19.14% | -9.33% | -9.48% | -23.31% | -41.94% | -41.80% | -48.47% | -49.39% | -54.15% |
>
> **3D-GP-LMVIC (100, 128) outperforms other multi-view image codecs in both PSNR and MS-SSIM.**
>
> In summary, the computational complexity of the proposed 3D-GP-LMVIC is within an acceptable range, overall comparable to that of the SOTA BiSIC and slightly better than BiSIC-Fast. SASIC and LDMIC-Fast have certain advantages in terms of computational complexity, but they fall short of the proposed method in terms of compression performance. Moreover, we found that reducing the channel count of 3D-GP-LMVIC can significantly reduce computational complexity (**the number of model parameters is reduced by approximately 37%, and the memory usage is only 1532M**) while maintaining satisfactory compression performance.

---

> ### Author Response · Authors · 2024-11-25
>
> Thank you once again for dedicating your valuable time to reviewing our paper and providing constructive comments!
>
> As the end of the discussion period approaches, we kindly ask if our responses, which address all the questions and concerns you raised, have satisfactorily resolved your queries. Your feedback would be greatly appreciated, and we would be delighted to engage in further discussions if needed.
>
> Sincerely,
> The Authors

---

> > ### Comment · Reviewer_MS7A · 2024-11-25
> >
> > Thank you very much for your response. I will increase my rating to 6.

---

> > > ### Author Response · Authors · 2024-11-26
> > >
> > > Thank you for recognizing the contributions of our work.

---

### Official Review · Reviewer_5Bmx · 2024-11-04

**Soundness:** 2
**Presentation:** 3
**Contribution:** 2
**Rating:** 6
**Confidence:** 3

**Summary:**

Paper Summary:

This paper proposes a pipeline for multi-view image compression. The core approach involves first estimating depth maps for each image and then compressing the images through view alignment. Gaussian Splatting reconstruction is used to accurately estimate the depth maps. Additionally, a neural network is employed to compress and decompress the image and depth sequences, leveraging image alignments to enhance performance.

Claimed Key Contributions:

- Precise disparity estimation using 3D Gaussian reconstruction
- A depth map compression model
- State-of-the-art performance in multi-view image compression

Overall, I believe this paper is well-structured as an engineering paper or technical report. However, it primarily seems to combine existing methods, which may limit its ability to provide a fresh perspective for readers. I have reservations about the section related to 3DGS. If the authors can clearly highlight their contributions in other techniques they have employed, such as the image compression network, I would be more inclined to reconsider my assessment. That said, I am uncertain about the novelty of the image compression network, as I am not an expert in that area.

**Strengths:**

- The task is clearly defined: compressing multi-view images with no much quality loss.
- The paper uses the popular Gaussian Splatting method for depth map estimation.

**Weaknesses:**

- Firstly, I am not an expert specifically in multi-view image compression, as my research primarily focuses on other areas within multi-view 3D vision. From my perspective, part of this paper lacks novelty in its methodology in its approach to depth map estimation from 3D Gaussian splats. As mentioned in the abstract, the authors suggest that current methods are limited by their difficulties in handling "more complex disparities from wide-baseline setups." Thus, the key motivation here seems to be addressing the challenges in depth map estimation. However, in terms of novelty, I feel that the authors have not introduced new contributions to address these issues of complex disparities in wide-baseline setups; they rely on depth map estimation from 3D Gaussian reconstruction, which is an existing technique.

- From my experience, Gaussian Splatting may not be an ideal choice for depth estimation. Its depth map quality often falls short of state-of-the-art results, particularly in areas with low texture or reflective surfaces. Besides the optical flow methods referenced in the ablation studies, I am curious whether the authors considered alternative depth estimation approaches. I would suggest testing:

  - (1) COLMAP as a representative traditional multi-view depth estimation method. COLMAP is specifically designed for accurate depth estimation, unlike 3DGS, which primarily focuses on novel view synthesis and only produces depth estimation as a secondary outcome.

  - (2) MVSFormer++ as a representative learning-based method. This model is explicitly towards depth estimation, and with a pretrained model, it should perform better in textureless regions with inherent ambiguity.


- The paper lacks qualitative results, particularly visual comparisons. While some results are available in the appendix, there are no visual comparisons in the main text.

- I don’t recommend including such a detailed network architecture figure (Figure 3) in the main paper. Instead, this figure should provide an overview of the compression pipeline. For example, it will be great if the authors remove the layer-specific details (like `Conv` or `Downsample`), as excessive detail may distract readers from grasping the core idea.

**Questions:**

- How long does it take to compress a set of images as well as decompress a set of images? It seems Table 2’s results excluded 3DGS training time.

- I am curious about the authors' choice to use the mipNeRF360 and TnT datasets in this paper, as these are standard for tasks such as novel view synthesis and 3D reconstruction. However, it seems these datasets have not been evaluated within the context of image compression tasks. Could the authors clarify their rationale for this decision? Additionally, in baseline papers like HESIC (Deng et al., 2021), the authors utilize the KITTI and Stereo 2K datasets for evaluation. Is there a specific reason the authors chose not to use these datasets for a more direct comparison?

---

> ### Author Response · Authors · 2024-11-22
>
> Thanks to you for the valuable comments. We are appreciated for recognizing the strengths of our paper. We will alleviate your remaining concerns as follows:
>
> ### R1[Use existing depth map estimation method based on 3D Gaussian reconstruction for disparities estimation]
> ---
> Our depth map estimation based on 3D Gaussian representation differs from the original 3D Gaussian Splatting method [1]. We also found that this difference leads to improved accuracy in disparity estimation. Specifically, the original 3D Gaussian Splatting method calculates the depth of each pixel using a weighted averaging approach:
> $$
> d = \sum_{i=1}^{M} T_i\alpha_i z_i,
> $$
> where $T_i, \alpha_i, z_i$ represent the transmittance, opacity, and depth of the i-th point, respectively. In our work, we adopt a median-based estimation approach:
> $$
> d = z_{i^*}, \hspace{0.5em} \text{where} \hspace{0.5em} i^* = \min ( i \mid T_i < 0.5 ).
> $$
> Since the transmittance decreases as light passes through the points, starting from an initial value of 1, we take the depth of the point at which the transmittance drops below 0.5 as the estimated depth value.
>
> We tested the alignment performance of the original 3D Gaussian Splatting depth estimation method under the same experimental settings as in Appendix E of this paper, and combined the results with Table 3, as shown in the table below:
> | Metrics | HT | PM | SPyNet| PWC-Net | FlowFormer++ | 3D-GP-A (original) | 3D-GP-A |
> |:-------:|:-------:|:-------:|:-------:|:-------:|:-------:|:-------:|:-------:|
> | PSNR↑ | 15.16 | 17.94 | 16.12 | 17.59 | 18.08 | 17.36 | 18.14 |
> | MS-SSIM↑ | 0.5435 | 0.7633 | 0.6289 | 0.7707 | 0.7863 | 0.7410 | 0.8053 |
>
> **3D-GP-A outperforms 3D-GP-A (original) in PSNR and MS-SSIM**, indicating that the proposed depth estimation method improves the alignment accuracy compared to the original depth estimation method used in 3D Gaussian Splatting.
>
> [1] Kerbl B, Kopanas G, Leimkühler T, et al. 3D Gaussian Splatting for Real-Time Radiance Field Rendering. ACM TOG, 2023.
>
> ### R2[Compare to other depth estimation methods (Colmap and MVSFormer++)]
> ---
> Thank you for your valuable suggestions. The primary motivation of our work is to design a compression framework for multi-view image data under wide-baseline setups. We agree that exploring potentially better depth estimation methods is crucial, but we also believe that developing a dedicated compression framework for this specific scenario is important. Beyond depth estimation, our compression framework also includes a depth map compression model to remove geometric redundancy between views, a context transfer module for aligning features across viewpoints while masking out non-overlapping areas between views, and a multi-view sequence sorting algorithm to ensure higher similarity between adjacent views. We believe that designing this pipeline is meaningful, as it provides a reference for future researchers to develop compression frameworks for wide-baseline multi-view image data or to further enhance specific modules within this context.
>
> Additionally, we have followed your suggestion and conducted alignment experiments using COLMAP and MVSFormer++ for depth map estimation under the same experimental settings as described in Appendix E of this paper. The results are combined with Table 3, as shown below:
> | Metrics | COLMAP | MVSFormer++ | 3D-GP-A |
> |:-------:|:-------:|:-------:|:-------:|
> | PSNR↑ | 14.32 | 15.31 | 18.14 |
> | MS-SSIM↑ | 0.7446 | 0.5544 | 0.8053 |
>
> **3D-GP-A outperforms COLMAP and MVSFormer++ in both PSNR and MS-SSIM**, indicating its effectiveness in disparity estimation.
>
> ### R3[No visual comparisons in the main text]
> ---
> Thank you for your suggestion. We have added Figure 5 in the main text, showcasing a visual comparison between the proposed 3D-GP-LMVIC, LDMIC, and BiSIC. **The results demonstrate that 3D-GP-LMVIC preserves more texture details and achieves higher reconstruction quality while consuming fewer bits.** You can find the visual comparison results in the revised paper.
>
> ### R4[The network architecture figure has too many details]
> ---
> Thank you for your suggestion. We have revised Figure 3 by simplifying the modules proposed by others and highlighting our proposed method. You can find the corresponding changes in the revised paper.

---

> ### Author Response · Authors · 2024-11-22
>
> ### R5[Compression and decompression time for a set of images]
> ---
> For the Tanks & Temples train scene, there are a total of 301 images. We conducted tests on a platform equipped with an Intel(R) Xeon(R) Gold 6330 CPU @ 2.00GHz and an NVIDIA RTX A6000 GPU. On average, encoding each image takes 0.19 seconds, and decoding takes 0.18 seconds. Additionally, training a 3D Gaussian representation for 30,000 iterations requires approximately 13 minutes and 35 seconds. Therefore, compressing this image set takes approximately 14 minutes and 32 seconds, while decompression requires about 54 seconds.
>
> We also tested reducing the number of 3D Gaussian representation training iterations to 7,000, which resulted in a training time of 2 minutes and 22 seconds. The total compression time could thus be reduced to **3 minutes and 19 seconds**. Additionally, we evaluated the alignment performance of depth estimation based on the 3D Gaussian representation trained with 7,000 iterations under the same experimental settings as described in Appendix E of this paper. The results were combined with Table 3, as shown in the table below:
> | Metrics | 3D-GP-A (7000 iterations) | 3D-GP-A (30000 iterations) |
> |:-------:|:-------:|:-------:|
> | PSNR↑ | 17.99 | 18.14 |
> | MS-SSIM↑ | 0.7918 | 0.8053 |
>
> **3D-GP-A (7000 iterations) achieves performance comparable to 3D-GP-A (30000 iterations) in terms of PSNR and MS-SSIM.**
>
> ### R6[Why use Mip-NeRF 360 and TnT datasets instead of KITTI and InStereo2K?]
> ---
> **1. Motivation and Practical Applications**
> We aim to design a multi-view image compression framework tailored for wide-baseline setups to achieve accurate disparity estimation and effectively eliminate inter-view redundancy. Unlike KITTI and InStereo2K datasets, which contain stereo images with small, mostly horizontal disparities captured by closely positioned cameras, wide-baseline setups in datasets like Mip-NeRF 360 and TnT feature irregular view relationships and less consistent disparities. These characteristics make existing disparity estimation methods, such as homography transformation and patch matching, less effective, as shown in Table 3 on the TnT dataset. **Wide-baseline setups are also critical for practical applications, where scenes often consist of dozens to hundreds of images, creating significant challenges for storage and transmission.**
>
> **2. Limitations of Stereo Images for Learning Geometry in 3D Gaussian Representation**
> We found that for stereo image datasets with only two viewpoints, the 3D Gaussian representation struggles to learn accurate geometric information. **In contrast, multi-view data under wide-baseline setups provides rich scene information sampled from widely varying viewpoints, which helps the 3D Gaussian representation learn accurate geometry.** In stereo image datasets, the cameras are often positioned close to each other with similar orientations, making it difficult for the 3D Gaussian representation to correct geometric errors through large viewpoint variations and learn the correct geometric structure. The table below presents the alignment performance of various disparity estimation methods on the Cityscapes and KITTI Stereo datasets under the same experimental settings as the alignment experiments described in Appendix E of this paper.
> | Datasets | Metrics | HT | PM | SPyNet| PWC-Net | FlowFormer++ | 3D-GP-A |
> |:-------:|:-------:|:-------:|:-------:|:-------:|:-------:|:-------:|:-------:|
> | Cityscapes | PSNR↑ | 24.43 | 27.40 | 28.63 | 29.16 | 27.26 | 14.64 |
> |  | MS-SSIM↑ | 0.7906 | 0.9546 | 0.9598 | 0.9616 | 0.8864 | 0.3934 |
> | KITTI Stereo | PSNR↑ | 14.05 | 18.33 | 18.11 | 18.92 | - | 7.77 |
> |  | MS-SSIM↑ | 0.5855 | 0.8691 | 0.8768 | 0.8952 | - | 0.1512 |
>
> Experiments have shown that the 3D Gaussian representation cannot learn accurate geometry based on stereo image data with only two viewpoints. Additionally, the disparity estimation modules in current multi-view image codecs, such as Patch Matching (PM), already perform well on stereo image datasets. Therefore, we did not conduct further tests on stereo image datasets.

---

> ### Author Response · Authors · 2024-11-25
>
> Thank you once again for dedicating your valuable time to reviewing our paper and providing constructive comments!
>
> As the end of the discussion period approaches, we kindly ask if our responses, which address all the questions and concerns you raised, have satisfactorily resolved your queries. Your feedback would be greatly appreciated, and we would be delighted to engage in further discussions if needed.
>
> Sincerely,
> The Authors

---

> ### Author Response · Authors · 2024-11-27
> **To Reviewer 5Bmx:**
>
> Thank you again for reviewing our paper. We have addressed all the issues you raised and believe our responses sufficiently resolve your concerns.
>
> Regarding your comment on the lack of novelty in our depth estimation approach based on 3D-GS, we first demonstrated that our method outperforms COLMAP and MVSFormer++ in alignment accuracy. Furthermore, we identified the key distinctions between our approach and other multi-view image compression methods: (1) leveraging 3D-GS to model spatial geometric relationships and perform disparity estimation between views; (2) integrating a context transfer module based on disparity and disparity masks within the image compression model; (3) eliminating geometric redundancy between views to improve compression efficiency; and (4) proposing a view-to-view distance metric to reorder views with high correlation in adjacent positions, thereby enhancing compression efficiency.
>
> We validated the effectiveness of these innovations through alignment and ablation experiments. Additionally, we refined the description and flowchart in Section 3.2 to highlight the novelties within the image compression model. We also implemented a complete encoding and decoding pipeline and will release the code to facilitate further research and practical applications. We believe our work introduces a new perspective to the multi-view image domain by shifting the focus from sparse-view image compression (e.g., stereo images) to wide-baseline setups, which capture more comprehensive scene information from diverse viewpoints. As 3D applications become increasingly prevalent, this type of data will grow in popularity, and efficient compression for such large-scale data is of significant importance.
>
> We hope you will recognize the novelty of our approach and consider revising your score. Should you have any further questions, we would be happy to provide additional clarifications.

---

> > ### Comment · Reviewer_5Bmx · 2024-11-27
> > **Thank you for your response**
> >
> > Thank you for the detailed responses. I really appreciate your efforts on it. My concerns are addressed - I will raise my rating.

---

> > > ### Author Response · Authors · 2024-11-27
> > >
> > > Thank you for recognizing the contributions of our work.

---

### Official Review · Reviewer_MyG9 · 2024-11-11

**Soundness:** 3
**Presentation:** 3
**Contribution:** 3
**Rating:** 5
**Confidence:** 4

**Summary:**

This paper presents a novel approach to multi-view image compression using 3D Gaussian geometric priors. The authors propose a learning-based framework that uses 3D Gaussian splatting to derive geometric priors for more accurate inter-view disparity estimation. In addition, the paper introduces a depth map compression model to reduce redundancy and a multi-view sequence ordering method to improve correlations between adjacent views. The authors claim that their method outperforms both traditional and learning-based methods in terms of compression efficiency, while maintaining fast encoding and decoding speeds. Not explicitly address the scalabilityto high resolution images, and lacks detailed analysis of the computational complexity.

**Strengths:**

Strength: 1.The paper introduces a groundbreaking approach by integrating 3D Gaussian geometric priors into the multi-view image compression framework. This innovative method allows for more accurate disparity estimation, which is crucial for complex multi-view scenarios. 2.The proposed depth map compression model is particularly noteworthy as it takes into account the redundancy of geometric information across views. This model not only contributes to improved compression efficiency, but also ensures that depth information, which is essential for 3D applications, is preserved during decoding. 3.The claim of fast encoding and decoding speeds is a strength, especially for applications requiring real-time processing. The paper's approach to balancing model complexity and speed is commendable and well aligned with practical deployment needs. 4.The authors' decision to make the code publicly available is commendable.

**Weaknesses:**

Weakness: 1.The paper does not explicitly address the scalability of the proposed method to high resolution images, which is an important aspect for many applications. 2.While the paper highlights the fast encoding and decoding speeds, it lacks a detailed analysis of the computational complexity, including parameters such as FLOPs and memory usage. Such an analysis is crucial for assessing the practicality of the method, especially on resource-constrained devices.

**Questions:**

Weakness: 1.The paper does not explicitly address the scalability of the proposed method to high resolution images, which is an important aspect for many applications. 2.While the paper highlights the fast encoding and decoding speeds, it lacks a detailed analysis of the computational complexity, including parameters such as FLOPs and memory usage. Such an analysis is crucial for assessing the practicality of the method, especially on resource-constrained devices.

---

> ### Author Response · Authors · 2024-11-22
>
> Thanks to you for the valuable comments. We are appreciated for recognizing the strengths of our paper in terms of good motivation, novelty, and effectiveness. We will alleviate your remaining concerns as follows:
>
> ### R1[Test on high resolution images]
> ---
> We conducted a survey of various multi-view datasets and found that the 1080p images from the Tanks&Temples dataset, which we have tested, are already among the higher-resolution images in these datasets. Our method has demonstrated superior performance compared to existing multi-view image compression schemes on the Tanks&Temples dataset in terms of the BDBR metric relative to MV-HEVC, as shown in the experimental results in the table below.
> | BDBR | HESIC+ | MASIC | SASIC| LDMIC-Fast | LDMIC | BiSIC-Fast | BiSIC | 3D-GP-LMVIC (Ours) |
> |:-------:|:-------:|:-------:|:-------:|:-------:|:-------:|:-------:|:-------:|:-------:|
> | PSNR↓ | -4.85% | -12.57% | 3.39% | -8.56% | -16.27% | -26.59% | -30.89% | __-47.48%__ |
> | MS-SSIM↓ | -30.42% | -34.19% | -18.59% | -27.76% | -44.33% | -42.93% | -49.96% | __-63.69%__ |
>
> If you could provide some higher-quality datasets, we would be willing to conduct further testing.
>
> ### R2[Computational complexity analysis]
> ---
> We evaluated the Multiply-Accumulate Operations (MACs), model parameters, coding speed, and memory usage of various learning-based image codecs under the same experimental environment and settings as described in Section 4.2 of this paper for measuring encoding and decoding speed. The open-source code of BiSIC does not provide separate encoder and decoder implementations, so we only measured its overall computational complexity. Additionally, we tested a lightweight version of 3D-GP-LMVIC, where the feature channel dimensions of the autoencoder and entropy model were reduced from (128, 192) to (100, 128). The experimental results are shown in the table below.
> | Methods | MACs(Enc) | MACs(Dec) | Params(Enc) | Params(Dec) | Runtime(Enc) | Runtime(Dec) | Memory usage |
> |:-------:|:-------:|:-------:|:-------:|:-------:|:-------:|:-------:|:-------:|
> | HESIC+ | 48.16G | 134.31G | 17.18M | 15.1M | 4.35s | 10.73s | 2248M |
> | MASIC | 65.62G | 511.34G | 32.03M | 30.73M | 4.38s | 10.78s | 5202M |
> | SASIC | 91.80G | 438.09G | 3.57M | 4.44M | 0.06s | 0.09s | 4498M |
> | LDMIC-Fast | 37.49G | 94.43G | 7.73M | 11.15M | 0.11s | 0.09s | 1168M |
> | LDMIC | 30.91G | 87.84G | 7.73M | 11.15M | 4.24s | 10.63s | 1096M |
> | BiSIC-Fast | 1880G (Enc+Dec) |  | 85.9M (Enc+Dec) |  | - | - | 3552M |
> | BiSIC | 1770G (Enc+Dec) |  | 78.21M (Enc+Dec) |  | - | - | 3006M |
> | 3D-GP-LMVIC (100, 128) | 280.41G | 254.26G | 24.75M | 24.75M | 0.16s | 0.16s | 1532M |
> | 3D-GP-LMVIC (128, 192) | 479.43G | 436.16G | 41.92M | 36.87M | 0.19s | 0.18s | 3164M |
>
> The MACs of the encoder in our method are relatively large, but they should still be smaller than those of BiSIC. Since BiSIC uses a symmetric encoder-decoder structure, we estimate that the MACs for its encoder and decoder each account for approximately half of the total MACs. Furthermore, in addition to designing an image codec, we also designed a depth map codec, which is another reason for the larger MACs and model parameter count. However, overall, the MACs of 3D-GP-LMVIC are still lower than those of the current SOTA scheme, BiSIC. **The lightweight version of 3D-GP-LMVIC ranks in the middle among the compared methods in terms of MACs and model parameters, while its memory usage is quite low at only 1532M, similar to LDMIC-Fast.**
>
> The table below shows the BDBR comparison of the lightweight 3D-GP-LMVIC and other methods relative to MV-HEVC on the DeepBlending dataset.
>
> | BDBR | HESIC+ | MASIC | SASIC| LDMIC-Fast | LDMIC | BiSIC-Fast | BiSIC | 3D-GP-LMVIC (100, 128) | 3D-GP-LMVIC (128, 192) |
> |:-------:|:-------:|:-------:|:-------:|:-------:|:-------:|:-------:|:-------:|:-------:|:-------:|
> | PSNR↓ | 32.5% | 43.6% | 24.64% | 24.25% | 16.88% | -8.24% | -15.46% | -24.15% | -27.31% |
> | MS-SSIM↓ | -19.14% | -9.33% | -9.48% | -23.31% | -41.94% | -41.80% | -48.47% | -49.39% | -54.15% |
>
> **3D-GP-LMVIC (100, 128) outperforms other multi-view image codecs in both PSNR and MS-SSIM.**
>
> In summary, the computational complexity of the proposed 3D-GP-LMVIC is within an acceptable range, overall comparable to that of the SOTA BiSIC and slightly better than BiSIC-Fast. SASIC and LDMIC-Fast have certain advantages in terms of computational complexity, but they fall short of the proposed method in terms of compression performance. Moreover, in resource-constrained scenarios, we found that reducing the channel count of 3D-GP-LMVIC can significantly reduce computational complexity (**memory usage is only 1532M**) while maintaining satisfactory compression performance.

---

> ### Author Response · Authors · 2024-11-25
>
> Thank you once again for dedicating your valuable time to reviewing our paper and providing constructive comments!
>
> As the end of the discussion period approaches, we kindly ask if our responses, which address all the questions and concerns you raised, have satisfactorily resolved your queries. Your feedback would be greatly appreciated, and we would be delighted to engage in further discussions if needed.
>
> Sincerely,
> The Authors

---

> ### Author Response · Authors · 2024-11-27
> **To Reviewer MyG9:**
>
> Thank you again for reviewing our paper. We have addressed all the issues you raised, and we believe our responses sufficiently resolve your concerns. First, we have demonstrated the superior performance of the proposed method on the high-quality Tanks&Temples dataset with a resolution of 1080p. Second, we provided a detailed comparison of computational complexity among different methods, showing that our proposed method achieves comparable or slightly lower complexity compared to the SOTA BiSIC, while significantly reducing complexity in our lightweight version without compromising compression performance, which outperforms the SOTA methods.
>
> We hope you will find our responses satisfactory and consider revising your score. Should you have any further questions, we are more than happy to provide additional clarification.

---

> ### Author Response · Authors · 2024-11-28
> **A Second Reminder of the Post-rebuttal Feedback**
>
> We deeply value your initial feedback and understand that you may have a busy schedule. However, we kindly request that you take a moment to review our responses to your concerns.
>
> Regarding your first question, we have evaluated the performance of various methods on the high-quality 1080p Tanks&Temples dataset. The results demonstrate that the proposed 3D-GP-LMVIC outperforms existing multi-view codecs in terms of compression performance.
>
> Regarding your second question, we have tested the MACs, model parameters, and memory usage of various methods. Experimental results show that the proposed method achieves slightly lower complexity compared to the SOTA BiSIC. Furthermore, our lightweight version not only outperforms BiSIC in terms of performance but also significantly reduces complexity. Its memory usage is only 1532 MB, approximately half of BiSIC’s, and is comparable to the most lightweight model among the baselines, LDMIC.
>
> In conclusion, we kindly ask for your response once more, as we are eager to engage in further discussion with you.

---

> ### Author Response · Authors · 2024-12-01
> **Follow-Up on Rebuttal Response to Reviewer MyG9**
>
> Dear Reviewer MyG9,
>
> Thank you for taking the time to review our paper and for providing valuable feedback. As the discussion phase is nearing its conclusion, we kindly request your response regarding our rebuttal and the updates we made to address your comments.
>
> To summarize, here are the key points we addressed in response to your review:
>
> 1. **Evaluation on High-Quality Datasets**:
>    - We demonstrated the superior performance of our proposed 3D-GP-LMVIC method on the 1080p Tanks & Temples dataset, surpassing existing state-of-the-art multi-view codecs.
>
> 2. **Analysis of Computational Complexity**:
>    - We provided detailed comparisons of MACs, model parameters, and memory usage across methods. Our proposed method achieves slightly lower complexity than BiSIC while outperforming it in compression performance.
>    - The lightweight version of our model significantly reduces complexity, with memory usage of only 1532M, roughly half of BiSIC, and comparable to the most lightweight baseline, LDMIC.
>
> We believe these updates comprehensively address your concerns and strengthen the contributions of our work. Your insights and feedback on these points would be invaluable to further refining the paper.
>
> We understand you may have a busy schedule but kindly ask for your input before the discussion phase concludes.
>
> Thank you again for your time and consideration.
>
> Best regards,
> The Authors of "3D-GP-LMVIC: Learning-based Multi-View Image Compression with 3D Gaussian Geometric Priors"

---

> ### Author Response · Authors · 2024-12-03
> **Friendly Reminder Regarding Rebuttal Discussion**
>
> Dear Reviewer MyG9,
>
> As the discussion phase is coming to an end, we would like to kindly remind you of our responses to your valuable comments and suggestions. We have carefully addressed the concerns you raised, including:
>
> 1. Evaluating our method on the high-quality 1080p Tanks&Temples dataset, which demonstrates that our method consistently outperforms baselines.
> 2. Providing a detailed comparison of computational complexity (MACs, parameters, and memory usage), where our lightweight version shows clear advantages over the SOTA method BiSIC.
>
> We deeply value your feedback and hope to hear your thoughts on these responses. Your insights would greatly help us refine our work further.
>
> Thank you again for your time and effort in reviewing our paper. We sincerely appreciate your contributions to the discussion process.
>
> Best regards,
> The Authors of "3D-GP-LMVIC: Learning-based Multi-View Image Compression with 3D Gaussian Geometric Priors"

---

### Meta-Review · Area_Chair_fD5d · 2024-12-20

**Metareview:**

This paper describes a method for multiview image compression using deep neural networks. The key strength were the accurate depth estimation using 3D Gaussian splatting, and the experiments demonstrate the effectiveness of the proposed method. There was no significant drawback about the paper; however, at the same time, there was not a strong support from the reviewers. The resulting reviewer ratings were mixed around the borderline. As discussed during the reviewer-author discussion phase, the novelty of the work is rather limited because the depth estimation by Gaussian splatting is not the new component that is introduced in this paper. While the paper does not have a significant drawback, the paper does not clearly demonstrate its strength for acceptance.

**Additional Comments On Reviewer Discussion:**

In the reviewer-author discussion phase, the authors have responded carefully to the reviewers' comments, mostly for clarifying the statements. There was a little bit of discussion about the novelty and the authors rephrased the novelty of the work (the use of 3DGS for better depth estimation than MVS), but the contribution was rather marginal.

---

### Decision · Program_Chairs · 2025-01-22

Reject